# Applications of Plasma-Activated Water in Dentistry: A Review

**DOI:** 10.3390/ijms23084131

**Published:** 2022-04-08

**Authors:** Noala Vicensoto Moreira Milhan, William Chiappim, Aline da Graça Sampaio, Mariana Raquel da Cruz Vegian, Rodrigo Sávio Pessoa, Cristiane Yumi Koga-Ito

**Affiliations:** 1Oral Biopathology Graduate Program, São José dos Campos Institute of Science & Technology, São Paulo State University, UNESP, São Paulo 12245-000, Brazil; aline.sampaio@unesp.br (A.d.G.S.); mary.rcv@hotmail.com (M.R.d.C.V.); cristiane.koga-ito@unesp.br (C.Y.K.-I.); 2Plasma and Processes Laboratory, Department of Physics, Aeronautics Institute of Technology, Praça Marechal Eduardo Gomes 50, São José dos Campos 12228-900, Brazil; chiappimjr@yahoo.com.br (W.C.); rspessoa@ita.br (R.S.P.); 3Department of Environment Engineering, São José dos Campos Institute of Science & Technology, São Paulo State University, UNESP, São Paulo 12247-016, Brazil

**Keywords:** plasma-activated water, plasma-treated water, atmospheric plasma, gliding arc discharge, DBD, dentistry, decontamination, oral cancer, tooth bleaching

## Abstract

The activation of water by non-thermal plasma creates a liquid with active constituents referred to as plasma-activated water (PAW). Due to its active constituents, PAW may play an important role in different fields, such as agriculture, the food industry and healthcare. Plasma liquid technology has received attention in recent years due to its versatility and good potential, mainly focused on different health care purposes. This interest has extended to dentistry, since the use of a plasma–liquid technology could bring clinical advantages, compared to direct application of non-thermal atmospheric pressure plasmas (NTAPPs). The aim of this paper is to discuss the applicability of PAW in different areas of dentistry, according to the published literature about NTAPPs and plasma–liquid technology. The direct and indirect application of NTAPPs are presented in the introduction. Posteriorly, the main reactors for generating PAW and its active constituents with a role in biomedical applications are specified, followed by a section that discusses, in detail, the use of PAW as a tool for different oral diseases.

## 1. Introduction

Plasma medicine is a multidisciplinary research field that investigates the uses of plasma in the healthcare field. Currently, non-thermal plasma technology is mainly focused on applications at atmospheric pressure, which is commonly referred to as non-thermal atmospheric pressure plasmas (NTAPPs). Direct applications of NTAPP have been used for decontamination of food [1,2] and food contact surfaces [1,3] in the food industry, in air purification [4] and as an antimicrobial agent in the medical and dentistry fields [5,6,7,8,9]. In 2021, NTAPP focused on plasma medicine reached a quarter of a century since the first published study [10]. Over time, in addition to the antimicrobial effects, other applications of NTAPP have been discovered in biomedical fields, such as the benefits for wound healing and cancer treatment [11,12].

NTAPPs can also be applied indirectly through the activation of water or liquids and through the treatment of contaminated or polluted water [13,14]. Their antimicrobial effect [15,16] and applicability in the treatment of cancer [17,18] and wound healing [19] have also been observed in the indirect modality of treatment. In recent years, plasma medicine has expanded the frontiers towards plasma dentistry. NTAPP has shown efficacy against oral microorganisms, in addition to anti-inflammatory properties, with possible application in cariology, endodontic, periodontics and oral oncology [20]. Besides, plasma-activated water (PAW) has also demonstrated potential application in dentistry fields [21,22].

It is important to emphasize the differences between both modalities of treatment: direct and indirect. The first implies that plasma is applied directly on a given substrate, such as wounds, skin, food etc., [23,24]. Conversely, in the indirect modality of NTAPPs, a given liquid is activated prior to the application to the substrate. To exemplify, Table 1 shows the main review articles published in the last three years on the applications of direct and indirect NTAPPs, in areas ranging from medicine, biomedicine, dentistry, agriculture and the food industry. It is worth mentioning that although the present work is focused on dentistry, the direct and indirect NTAPPs are of great importance for agriculture and the food industry. This wide range of applications of direct and indirect NTAPPs is schematically presented in Figure 1. Figure 1a highlights the applicability of direct NTAPPs that cover different human and food healthcare areas. Figure 1b illustrates the applications of NTAPPs through the modality of generating plasma-activated liquid (PAL) or PAW, which are later applied to substrates. As described, PAW and PAL studies are focused on different purposes, which shows the effort of many scientists to improve and make plasma technology more accessible.

To understand the mechanisms of action related to NTAPPs, it is essential to know that the plasma is a partially ionized gas consisting of particles as electrons, ions, metastable species, radiation as ultraviolet (UV), visible (VIS), the electromagnetic field and reactive species. It is worth highlighting that the term reactive species is commonly used for free radicals and reactive oxygen species (ROS) and reactive nitrogen species (RNS). It is highly reactive due to the presence of unpaired valence electrons or non-static bonds on its structure. Despite the various components produced in plasma, reactive oxygen and nitrogen species (RONS) play a fundamental role in plasma medicine [32,33]. RONS generated in NTAPP are divided into short-lived and long-lived species. Hydroxyl radical (OH^−^), delta oxygen singlet (^1^O_2_) and superoxide anion (O_2_^−^) are examples of short-lived species, which last from a few seconds to minutes [34]. In contrast, hydrogen peroxide (H_2_O_2_), nitrite (NO_2_), nitrate (NO_3_^−^), nitrous acid (HNO_2_) and ozone (O_3_) belong to the groups of long-lived species [35,36]. When NTAPP is used in an indirect mode, i.e., in liquid or water activation, most RONS found in the liquid phase are long-lived ones, generated from the gas phase of the plasma and the plasma–liquid interface interaction. The generation of RONS also occurs through the primary transformation of reactive species generated in the liquid [37]. Therefore, water or liquid exposure to NTAPP induces active constituents, called RONS, which are useful for biomedical applications [21].

Due to the growing interest of the scientific community in plasma technology applied to dentistry, a review article is presented here to serve as a quick guide for dentists, physicians, physicists, engineers and health professionals in general. This review aims to discuss the applicability of PAW in dentistry, which, as can be seen in Table 1, is a topic review with little or no exploration. Thus, based on the published literature, the findings of PAW applied to dentistry are presented, as well as the future perspectives for the area considering the main results related to direct and indirect NTAPP for biomedical applications.

**Table 1 ijms-23-04131-t001:** Summary of the main review articles published in the last three years on applications of direct and indirect NTAPPs.

Approach	NTAPP Modalities	Publication Year/Reference
Molecular Mechanisms of the Efficacy of NTAPP in Cancer Treatment	Direct	2020 [38]
A Powerful Tool for Modern Medicine	Direct	2020 [11]
The New Frontier in Low Temperature Plasma Applications	Direct	2020 [39]
Atmospheric Cold Plasma Treatment in Fruit Juices	Direct	2020 [40]
Chemical, Physical and Physiological Quality Attributes of Fruit and Vegetables Induced by Cold Plasma Treatment	Direct	2020 [41]
Cold Plasma as a New Hope in the Field of Virus Inactivation	Direct	2020 [42]
The Effects of Plasma on Plant Growth, Development and Sustainability	Direct	2020 [43]
Cold Plasma for the Control of Biofilms in Food Industry	Direct	2020 [44]
Potential of Cold Plasma Technology in Ensuring the Safety of Foods and Agricultural Produce	Direct	2020 [45]
Plasma Agriculture from Laboratory to Farm	Direct	2020 [46]
Aurora Borealis in Dentistry	Direct	2021 [47]
Applications of Cold Atmospheric Pressure Plasma in Dentistry	Direct	2021 [20]
Cold Atmospheric Pressure Plasma Technology in Medicine, Agriculture and Food Industry	Direct	2021 [48]
The Antimicrobial Effect of Cold Atmospheric Plasma against Dental Pathogens	Direct	2021 [49]
Influence of Atmospheric Cold Plasma on Spore Inactivation	Direct	2021 [50]
Plasma-Assisted Agriculture: History, Presence and Prospects	Direct	2021 [51]
Improving Seed Germination by Cold Atmospheric Plasma	Direct	2022 [52]
Reactive Nitrogen Species in Plasma-Activated Water	Indirect	2020 [53]
Recent Advances in Plasma-Activated Water for Food Safety	Indirect	2022 [54]
Influence of Plasma-Activated Water on Physical and Physical–Chemical Soil Properties	Indirect	2020 [55]
PAW Triggers Plant Defense Responses	Indirect	2020 [56]
PAW Generation, Origin of Reactive Species and Biological Applications	Indirect	2020 [36]
Interactions of Plasma-Activated Water with Biofilms	Indirect	2020 [57]
A Comprehensive Review of PAW for Enhanced Food Safety and Quality	Indirect	2021 [58]
Applications of PAW in the Food Industry	Indirect	2020 [59]
PAW for Cancer Treatment: Positives, Potentials and Problems of Clinical Translation	Indirect	2020 [60]
Review on Discharge Plasma for Water Treatment	Indirect	2020 [61]
PAW on Microbial Growth and Storage Quality of Fresh-cut Apple	Indirect	2020 [62]
PAW Production and its Application in Agriculture	Indirect	2021 [63]
Diagnostic Analysis of Reactive Species in PAW: Current Advances and Outlooks	Indirect	2021 [64]
PAW from DBD as a source of Nitrogen for Agriculture	Indirect	2021 [65]
PAW, Hydrogen Peroxide and Nitrates on Lettuce Growth	Indirect	2021 [66]

Therefore, this review is divided as follows; Section 2 shows the main reactors used for PAW/PAL and plasma-treated water (PTW). Another essential point demonstrated in this section is the formation of RONS, the fundamental species for plasma medicine. Section 3 explores the main text of this review, focusing on PAW applied to different areas of dentistry. In this context, the following topics are explored: decontamination of dental devices, treatment of oral infectious diseases, anti-inflammatory properties and wound healing, anti-cancer therapy and tooth bleaching. Finally, Section 4 presents the conclusion of the work.

## 2. Plasma-Activated and Plasma-Treated Water

In the literature, the terms treatment and activation are often misused. To avoid doubt, treatment is defined as a practical means or refinements used to combat/mitigate a problem. In the specific case of water treated by plasma, the term treated refers to the process of elimination or complete mineralization in wastewater of synthetic dyes, pharmaceuticals products and pathogenic bacteria, among other pollutants. Therefore, plasma water treatment is commonly used to purify or decontaminate small or large amounts of water. In contrast, activation is understood as increased activity, becoming active, boosting, accelerating, or intensifying some specific property. In this case, the non-pollutant water exposed to the plasma becomes activated, i.e., it obtains new properties. Unlike treatment, activation is carried out in small amounts of water (between 1 mL to 1000 mL) [36,53,66], but the reactors used are the same, with a slight modification in both cases. It is important to note that deionized water, distilled water, filtered water and potable tap water, i.e., pure water without pollutants, are usually used for activation.

### 2.1. Main Reactors of Producing

Two main types of plasma reactors are used in PAW generation: dielectric barrier discharges (DBDs) and plasma jets (PJs). Both are non-thermal plasmas, i.e., they are NTAPP and have a wide range of types [65,67]. DBDs for PAW generation are considered indirect sources of plasma, as the plasma is mainly produced between the reactor electrodes without any contact with the water, taking advantage of the interactions of the gas with the liquid. Some reactive plasma species can reach the water surface through ionizing wave mechanisms using electric field propagation, convection through airflow and diffusion [65]. In contrast, the PJs used for PAW generation are considered direct sources of plasma, i.e., there is direct contact between the plasma and the water and the water can act as a counter electrode so that the discharge current can flow through the liquid. The PJs are the most widely used reactors, as some of them generate a stable volume of plasma that is controllable without the confinement between the electrodes, as in the case of the DBDs [67]. In both reactors, the formation of RONS occurs at the interface between the gaseous and liquid phases and/or within the liquid, which drastically changes the concentration of the RONS constituents [36]. In contrast, in the treatment of water by plasma, the reactors can be immersed in the treated liquid [68,69]. Therefore, in this case, both DBD reactors and PJ reactors can be used.

It is important to note that NTAPPs operate at a high voltage. As reported in the literature, these voltages range from 1 to 50 kV, with operating frequencies that can start in the tens to thousands of Hz (kHz) and powers that generally do not exceed values greater than 10 W. Commonly used working gases are helium (He), argon (Ar), oxygen (O_2_), nitrogen (N_2_), air or a mixture of these gases. Working gases are used with flow rates ranging from 1 to 30 L/min [36,53,54,55,56,57,58]. Indeed, there is a range of reactors used for water and liquid activation and, every day, a new article is published with new reactors that have minor changes. Therefore, a dedicated review article would be needed to demonstrate the NTAPP generation reactors and their characteristics, but the focus of the present work is not that. Thus, below we show some reactors used to generate PAW.

As seen in Figure 2, discharges are used directly into the water and on the surface of the water, which considerably affects the chemical composition of PAW. This difference in chemical composition is basically due to the differences between the rupture forces in the gas phase (discharge on the surface) and the water (discharge inside) [70,71]. However, as reported in the literature, the most applied systems are those that operate with plasma discharge in contact with the water, i.e., PJ and DBD as a plasma source [72,73]. As demonstrated in the next section, these plasmas deliver the RONS from the plasma gas to the liquid phase more efficiently. 

An important question is whether there is commercial equipment that can be applied in clinical practice. Recently, Andrasch et al. [72], Pemen et al. [73] and Schnabel et al. [75] developed pilot units with potential for practical applications. Andrasch et al. [72] and Schnabel et al. [75] obtained a PAW production rate of 1 L/min. Pemen et al. [73] obtained 0.5 L/min. However, these devices cannot meet the application requirements in dentistry and medicine, which are low pH and high concentrations of RONS. On the other hand, in agriculture and in the food industry, the production of millions of liters of PAW at a low cost is required. Therefore, it can be said that PAW-generating equipment is still in the pilot phase and has great commercialization potential in the coming years.

### 2.2. Origin of Reactive Oxygen and Nitrogen Species

The RONS induced in PAW are dependent on many different parameters such as (i) the composition of the solution, (ii) the distance between the plasma and the liquid surface, (iii) the types of power supply used in the plasma generation, (iv) electrode configuration, (v) applied voltage, (vi) voltage polarity, (vii) water volume, (viii) gas type and flow, among other parameters [31,76,77,78]. It is also important to highlight that both the chemistry and the reaction processes of PAW produced with the reactor in a few centimeters of water are different from those made in the generation of PAW with the reactors submerged in water. The gas–liquid system generated for electrical discharges placed in a few centimeters of water is more interesting, as they can cause post-electrical discharge reactions capable of forming other long-lived reactive species [59,70]. For example, when using compressed air as the plasma-generating gas for water activation, the gas–liquid interface produces numerous short-lived species, ranging from hydroxyl radicals, superoxide and nitrous oxide to the generation of atomic oxygen and nitrogen [36]. Short-lived species later generate long-lived species, such as nitrites, nitrates and hydrogen peroxide (see Figure 3).

With the significant increase in PAW applications, it is necessary to understand which RONS can be formed and how they are formed, in addition to the quantification of their concentrations. In Appendix A, the generation and recombination mechanisms of the main plasma-induced long-lived reactive species are described in water activation processes that play a crucial role in healthcare applications.

## 3. PAW Applied in Dentistry

### 3.1. Decontamination of Dental Devices

The surfaces of medical and dental devices are constantly exposed to different microorganisms during procedures, working as a reservoir of pathogens and a potential source of contamination for the patients [79]. To avoid the risk of contamination, different methods and chemical products may be used to control pathogenic or nonpathogenic microorganisms. However, limitations such as ineffective sterilization, low penetration, incompatibility with materials that are sensitive to heat and corrosion, residue release, toxicity to the environment and individuals [80,81] and an inability to penetrate biofilm cell structures [82] support a constant search for new methods. In addition to being useful in the inactivation of microorganisms when applied over, inside, or touching the water surface [83], NTAPPs are also efficient against prion proteins [84], which are resistant to traditional cleaning methods [85].

The direct application of NTAPPs for the disinfection of heat-sensitive materials has been explored in the literature [86,87]. The disinfection of endodontic devices [26], silicone, diamond dental drills [88], metals [89], titanium alloy surfaces [90] and titanium disc surfaces [91] were analyzed with satisfactory results. The effectiveness of biological decontamination by NTAPPs depends on some parameters such as equipment configuration (frequency, power), gas (type and flow), the geometry of the analyzed material, distance, exposure time and also the position and design of the device [80,86,92]. Depending on the geometry of the material, multiple exposures may be required to eliminate the microorganisms.

As an alternative method, solutions activated with plasma, such as PAW, have also been evaluated for disinfection [93]. In fact, this in an interesting field of investigation as the possibility of disinfecting a device with a simple washing seems easier, compared to multiple exposures that may be needed to decontaminate a device/biomaterial with a complex geometry. It was previously observed that microbicide action may be related to the main reactive species and also the low pH acquired during aqueous plasma activation [94]. The use of PAW for the sterilization of medical devices was previously suggested, considering its antimicrobial properties [95]. One of the main advantages of this technology is that plasma-activated liquids may keep the antimicrobial effect over one month after PAW generation, when stored at a minimum of −80 °C [96,97]. However, the antimicrobial activity may be reduced, depending on the storage temperature, by decreasing the number of reactive species. Inactivation of *S. aureus* was observed after 20 min of treatment with PAW stored at −80 °C. However, this potential decreased significantly when plasma-activated distilled water was stored at −20 °C for 1, 7, 15 and 30 days [96]. Similarly, another study observed that 60 min of exposure to PAW stored at –80 or –150 °C may promote microbial inactivation of *S. aureus* and *E. coli*, even after six months of storage. On the other hand, temperatures equal to or higher than −16 °C showed reduced antimicrobial properties in short or long periods of storage [97]. In addition to the possibility of disinfecting dental materials in the easiest way, the conservation properties of PAW indicate another advantage of using this technology compared to direct NTAPP, since it would considerably reduce the demand for using the plasma device.

The decontamination of dental unit waterline system tubes (DUWLs) is a challenge for dentistry due to the risk of cross-contamination [98] and also the limitation of some traditional disinfectants, which may be toxic [79,99]. The reduction in the viability of mature *Enterococcus faecalis* biofilms formed in DUWLs for 5 days after PAW treatment was detected. For this, distilled water was activated for 3 min by a continuous plasma jet of compressed air gas. A significant reduction in viable cells was observed after 1–3 min of treatment. Treatment for up to 5 min led to a 100% reduction. Moreover, 3 min of contact showed similar effects to 1% H_2_O_2_ and 10 mg/L NaOCl; antimicrobial agents that are routinely used. The low pH (2.21) and the presence of NO, OH, NO_3_ and H+ species probably contributed together to the bactericidal effect [99]. Further investigations on PAW with this type of device and other microorganisms would be interesting.

The disinfection of stainless steel and polyethylene substrates by PAW has also been explored [100]. It is important to highlight that these materials present a wide application in the dental fields [101,102]. For these tests, 20 mL of sterile distilled water was activated for 5 min by non-thermal GlidArc plasma formed by moist air gas. After 30 min of exposure to PAW, microbial reduction both in planktonic and biofilm forms, were observed on Gram-positive and Gram-negative bacteria, as well as in yeasts. More than a 5 log reduction in viable cells was observed for bacterial biofilms, while there was a decrease of approximately 3 log in yeasts. Advantageously, the disinfection of the solid materials with PAW did not show damage to the materials, especially to stainless steel, which was free of corrosive signs [100].

Resin based materials have an important applicability in dentistry [103]. Sterilization by PAW of a duodenoscope coated with a polymer resin was also previously analyzed. Distilled water (300 mL) was activated for 10 min, with a discharge of GlidArc plasma operated with air gas. The treatment produced an acidified water of pH 2.78 that reduced the viable cells of *Escherichia coli* and *Acinetobacter baumannii* after 20 min of exposure, without any damage to the surface and composition of the equipment. The decontamination effect was also observed for *Klebsiella pneumoniae* and *Pseudomonas aeruginosa* after 30 min of PAW exposure [104].

Taken together, the studies with PAW have shown good decontamination properties of materials that are widely used in dentistry, such as stainless steel, polyethylene and polymer resin. These results bring insights for a new sterilization method in the dental field. Despite that, new studies with other materials and mostly with polymicrobial biofilms, which are closer to the clinical environment [105], would be interesting to explore this application.

### 3.2. Treatment of Oral Infectious Diseases

The use of PAW in the treatment of oral infectious diseases is a promising field of investigation. Several groups of microorganisms are involved in the etiopathogenesis and progression of the main infectious diseases that affect the oral cavity, such as caries, periodontitis and candidiasis. Considering the antimicrobial properties of NTAPPs, the control of these microorganisms by PAW or other plasma-activated solutions has also been investigated as an alternative treatment to traditional therapies that have their limitations.

Biological, behavioral, psychosocial and environmental factors are related to the development of caries [106]. The imbalance between demineralization and remineralization, caused by fluctuations in pH, promotes tooth decay [107]. The metabolic activity of dental biofilm, formed by microorganisms embedded in a matrix of extracellular polymeric substances adhered to the teeth surface, is responsible for these pH fluctuations, especially with the intake of a sugar-rich diet. The biofilm continues to grow progressively if undisturbed. In its composition, there are many groups of microorganisms and some of them are especially involved in the carious process [108].

*Streptococcus mutans* and *Lactobacillus* spp. are considered the main cariogenic bacteria responsible for producing acid and, consequently, the demineralization of the tooth structure [109,110]. Interestingly, PAL has shown to be effective against cariogenic microorganisms [22,111]. A previous study observed that PBS or a saline solution activated with a non-thermal plasma of argon (Ar) and oxygen (O_2_) for 5 min are able to reduce the number of *S. mutans* viable cells. This reduction was verified in both planktonic and biofilm forms of *S. mutans* after 1 h of treatment. The authors further reported that the activated liquids were not cytotoxic to fibroblasts [111].

Modifications in the composition of the oral biofilm may be observed due to dietary habits, type of dentition (primary or secondary) and even with the disease progression. Although *S. mutans* and *Lactobacillus spp.* are considered the most important microogasnisms related to dental caries, previous findings demonstrated that *Actinomyces spp*. may be associated with the disease progression in root caries [108]. A previous study evaluated the role of distilled water activated by a plasma jet of Ar and O2 (98% and 2%, *v*/*v*, respectively) for 20 min, in the reduction in *S. mutans*, *Porphyromonas gingivalis* and *Actinomyces viscosus*. The treatment that was performed from 0 to 120 s reduced the viability of all microbial species. A significant reduction in *A. viscosus* was observed within 40 s while *S. mutans* achieved a similar reduction after 60 s of treatment [22].

*Candida albicans* is another microorganism that may be found in the carious dentin of active root carious lesions and some authors have suggested that this fungus might play a role in the progression of the disease [112]. This species was detected on the biofilm in cases of childhood caries [113] and it was associated with an increase in plaque glucosyltransferase (Gtf) enzyme activity, a virulence factor associated with caries, in children with early carious lesions [114]. In addition to its possible role in dental caries, *C. albicans* is the main species related to oral candidiasis, the most common oral fungal disease. Local and systemic factors such as impaired salivary gland function, inhaled steroids, dentures, oral cancer/leukoplakia, broad-spectrum antibiotics, immunosuppressive drugs and conditions, nutritional deficiencies, diabetes, smoking and Cushing’s syndrome may contribute to oral candidiasis [115]. Previous studies demonstrated that the direct application of NTAPPs is effective against *C. albicans* [116,117], both in planktonic and biofilm forms. Investigations into the effects of PAW on *C. albicans* have also been conducted with different methodologies and findings. A reduction in *C. albicans* viability after 5 min of treatment with distilled water, activated by a dielectric barrier discharge (DBD) with atmospheric air, for 5 and 10 min, was reported. The authors attributed the antimicrobial action to the higher concentration of NO^−^_3_ and lower concentrations of NO^−^_2_ and H_2_O_2_ in PAW [118]. Antimicrobial effects against *C. albicans* was also reported in hydrogels constituted by plasma-activated deionized water, after 24, 48 and 72 h of contact time. An increase in the inhibition zones of the microorganism was observed in longer exposure times, with the best result after 30 min. Hydroxyl radical and NO^−^_3_ were suggested to be the main components with antifungal activity [119]. On the contrary, one study reported no effect of plasma-activated tap water for 10 and 30 min on *C. albicans* planktonic cells. The water was activated with an atmospheric air plasma, generated by a forward vortex flow reactor (FVFR), for 30 min [16].

In the context of cariology, it is also important to emphasize the positive property of NTAPPs in adhesive restorations. Plasma exposure generates the deposition of free radicals and ions on the tooth substrate, changing the surface proteins of dentin, which has led to an increased bond strength in adhesive restorations. The enhance in bone bond strength avoids microleakage and consequently prevents secondary caries [47]. It is not known whether PAW can also increase the bond strength, improving the restoration performance. This process could also probably occur with plasma–liquid technology due to the action of the reactive species on the surface of dentin.

In the evolution of the carious process, traumatic injuries and cracks allow that pathogens and their products to pass through the dentin and reach the pulp. The pulpal infection frequently progresses to necrosis of the tissue and the infection may spread to the apex of the tooth, promoting periapical disease [120]. *Enterococcus faecalis* is commonly found in primary and secondary endodontic infections [120,121,122]. This microorganism expresses virulence factors and resistance mechanisms that favor its presence in the root canal and consequently can lead to endodontic therapy failure [120,123]. Some studies have investigated the action of PAW on *E. faecalis* [99,124], which is interesting for dentistry as it could be used as an antimicrobial irrigator in endodontic treatments. Considering the particularities of root canals, such as the presence of accessory canals, the antimicrobial irrigation with PAW would be more useful than the direct application of NTAPPs. As mentioned, a reduction in *E. faecalis* viable cells in 5-day DUWLs biofilms was observed after 1–3 min of treatment with previously activated water. The treatment of the *E. faecalis* suspensions, in deionized water, was also performed with satisfactory results. The bacterial suspension was exposed to microjet plasma formed by atmospheric air for 10 to 90 s. The antimicrobial effect occurred progressively after 45, 60 and 90 min of treatment, while inhibitory effects on the biofilm formation was detected even in shorter exposure times (10, 20 and 30 s) [124]. Another microorganism that may be isolated from root canals, though not often, is *Escherichia coli* [120]. It was demonstrated that the planktonic treatment with PAW significantly reduced the number of *E. coli* colonies after 10 min of exposure [16]. In this same study, 10 and 30 min of PAW contact was effective against *Staphylococcus aureus*, a bacteria commonly found in chronic osteomyelitis of the jaw, in association with anaerobic pathogens [125].

The development and progression of periodontal disease is related to specific groups of Gram-negative bacteria, the so-called periodontopathogens. The transition from healthy periodontium to periodontitis is related to three important factors: the polymicrobial synergy, the dysbiotic microbiota and a susceptible host [126]. The multifactorial etiology of periodontal disease contributes to the difficulty in the treatment and alternative therapies have been investigated [127]. *Porphyromonas gingivalis*, *Treponema denticola*, *Tannerella forsythia* and *Aggregatibacter actinomycetemcomitans* have been considered important periodontopathogens [128]. A progressive inhibition of *P. gingivalis* in planktonic and biofilm forms was previously observed after 1, 3, 5 or 7 min of NTAPP exposure. Moreover, improved periodontal tissue recovery was obtained after 5 min of exposure, proportionally to the number of applications [129]. Interestling, a previous study demonstrated that PAW may also be effective against *P. gingivalis.* A reduction of 5-log in planktonic bacteria was observed after 20 s of exposure [22]. Considering the particularities of subgingival biofilm and its relationship with the progression of the disease, the treatment with PAW would be even more interesting since it would be able to reach areas of restricted access, such as the subgingival sites.

The mechanisms of action suggested in all of these studies, evaluating the antimicrobial properties of PAW, mainly involved the reactive oxygen and nitrogen species (RONS) produced in the solutions by plasma activation. Different biological effects could be observed in each study, which is probably related to differences in methodology and groups of microorganisms. Further in vitro and in vivo investigations are needed to standardize the best parameters for each solution and microorganism. Considering the potential antimicrobial effects observed in these studies, they could probably contribute to the treatment of oral infectious diseases.

### 3.3. Anti-Inflammatory Properties and Wound Healing

An important feature of non-thermal plasma is the possibility of tissue antisepsis without causing damage, which makes NTAPPs a good alternative treatment for infectious diseases. This selectivity probably occurs due to biochemical, metabolic and cell cycle differences between eukaryotes and prokaryotes and also the surface/volume ratio of mammalian cells, that is higher compared to bacterial and fungal ones [130]. It was demonstrated that no important side effect occurs after oral application of NTAPP on the mucosa of mice, in short-term experiments [131]. A previous study investigated the effect of PAW intake in mice after 90 days of administration, as its use in dentistry may lead to accidental ingestion. The mineral composition and surface micro-morphology of vital mouse teeth after long-term exposure, as well as local and systemic toxicity, were evaluated. The authors observed that there were not significant changes in the mineral composition and surface micro-morphology of the teeth. Moreover, the long-term exposure was not toxic to the tongue, oral mucosa, sublingual glands or other body organs, which presented normal structure and physiology [132]. In addition to not being harmful to mammalian cells, NTAPPs have been demonstrated to decrease inflammation and contribute to tissue repair [133].

Studies on skin inflammatory diseases such as allergic contact dermatitis and atopic dermatitis have shown anti-inflammatory effects of non-thermal plasma [134,135,136]. These effects have also been observed in oral studies. The treatment of oral candidiasis in mice showed a low occurrence of inflammatory alterations. After plasma exposure, the cell inflammatory infiltrate was predominantly mononuclear and macrophage-rich, with scarce polymorphonuclear cells [116]. Additionally, a study evaluating the role of NTAPP as an adjuvant therapy for the treatment of periodontitis induced in rats, observed that the expression of inflammatory-related cytokines such as TNF-α and IL-1β decreased significantly in the group where NTAPP was used as an adjuvant approach, while the level of the anti-inflammatory cytokine IL-10 showed a significant increase [137].

The behavior mast cells and keratinocyte cell line (HaCat) after the contact with non-thermal plasma-activated medium has been analyzed. Interestingly, the plasma-activated liquid prevented an enhancement of the pro-inflammatory genes and cytokines TNF-α, IL-6 and IL-13 in activated mast cells, by inhibiting the NF-κB signaling pathway. The activation of NF-κB by TNF-α/IFN-γ was also inhibited in HaCat cells suggesting that this treatment could be effective against, not only acute, but also chronic inflammation [136]. Similarly, in another study the pro-inflammatory responses of HaCat, activated by TNF-α/IFN-γ or LPS, was also suppressed by PAL. Moreover, STAT3, which is an important pathway for Th17 cell activation, was inhibited by PAL in IL-6-stimulated HaCaT [135]. In this way, plasma-activated liquids have shown to act in different inflammatory signaling pathways of keratinocytes. These findings are important for dentistry as NF-κB e STAT3 pathways are involved in the etiopathogenesis of periodontitis [138]. Moreover, these pathways also play a role in oral candidiasis as mucosal candidiasis promotes NF-κB activation [139] and STAT3 signaling is related to IL-17-mediated immunity in oral mucosal candidiasis [140].

The possibility of treating autoimmune skin diseases by direct application of plasma or plasma-activated liquids may also be interesting for dentists, who also have to deal with some autoimmune conditions, such as oral lichen planus [141], pemphigus and mucous membrane pemphigoid [142]. The effects of NTAPP on oral lichen planus (OLP) were previously investigated [143]. For this, biopsies from healthy and OLP areas were performed, followed by the application of NTAPP in the ex-vivo tissues for 3 min. From these lesions, 24 were reticular, 3 erosive and 1 atrophic. The treatment decreased the infiltration of T-cells in OLP, compared with healthy samples. Additionally, the levels of IL1β, IL2, IL10 and GM-CSF decreased significantly after the treatment and a tendency to decrease other inflammatory markers was observed, suggesting an immunomodulatory role of NTAPPs in OLP. The authors also presented a clinical report from a 73-year-old man suffering from erosive OLP. The treatment consisted of 5 min of application, two to three times per week (12 sessions). It promoted relief of the burning sensation after four sessions. During the treatment, the oral inflammation decreased and the ulcerated area of the lesion healed almost totally [143]. Considering that the erosive presentation is usually symptomatic, requiring treatment with topic steroids and sometimes systemic ones [141], clinical studies evaluating a large number of these cases are welcome. The efficiency of PAW should also be evaluated, since washing with a plasma-activated liquid could reach the entire area of OLP without the need for direct application at several areas of the lesion.

The exposure of diabetic animals to NTAPPs has also shown anti-inflammatory properties [144,145] and improvement in wound healing [146], which brings perspectives for dentistry, especially considering the proven relationship between diabetes and periodontal disease. Severe periodontal destruction is usually observed in diabetic patients, while poor glycemic control is more common in diabetic people who also have periodontal disease [147]. Interestingly, diabetic mice treated with NTAPP showed a decrease in oxidative stress biomarkers, advanced glycation end products (AGEs) and inflammatory cytokines, such as IL-1, IL-6 and TNF-α [145]. It has been suggested that increased accumulation of AGEs and their interaction with specific receptors (RAGE) in diabetic gingival tissue could promote the hyperproduction of proinflammatory cytokines, as well as vascular alterations and a loss of tissue integrity, contributing to the worsening of periodontitis [148]. Thus, the adjunct treatment for diabetes mellitus with plasma modalities could possibly act indirectly and positively in periodontitis and in other inflammatory oral diseases. The direct action of NTAPPs or PAW in periodontal disease should also be considered, since it is a multifactorial condition in which the microbial biofilm activates the immune system with the production of proinflammatory cytokines and, consequently, tissue loss [149]. The selective role of plasma could be useful in both, in the elimination of periodontopathogenic microorganisms and also in the gingival tissue, decreasing the inflammatory process. Considering the generalized subtype of chronic periodontitis, treatment with PAW could be clinically easier, reaching different affected areas in a single use.

The anti-inflammatory and microbicide functions of NTAPPs have been demonstrated to favor tissue repair [133]. Clinical trials evaluating NTAPPs have already been performed, in which this technique was considered safe, painless and effective against bacterial load [150,151]. Solutions activated with plasma, such as medium, saline and water have also shown good results in vitro and in vivo concerning the wound healing [19,133,152]. Cell proliferation and migration were observed in human keratinocytes exposed to 15 s of medium activated with Helium-and-Argon (He/Ar)-generated NTAPP [133], which could favor the re-epithelialization of wounds on the skin and also on the oral mucosa that has keratinocytes in the epithelial composition. There is no study evaluating the effect of PAW or PAL on mouth wound healing, although two studies carried out in rats and mice have observed a tendency to improve periodontal tissue loss after the experimental treatment of periodontitis with NTAPPs [129,137]. Moreover, wound healing of some infected ulcerated areas of advanced oral squamous cell carcinoma was observed after NTAPP exposure [153]. Considering that wound repair is important for different modalities of dentistry such as oral surgery, periodontics, oral pathology and implantology, PAW could be an option to accelerate the healing after oral diseases or oral surgeries.

NTAPPs were previously suggested as a good possibility for oral surgery because NTAPP was tested with osteoblast-like cells (MG-63), leading to cell proliferation and in vitro wound closure [154]. Oral implant modification with NTAPPs has also been suggested since this treatment may enhance the roughness and wettability of the implant surface, thus improving the cell adhesion and consequently the osseointegration [47]. These results with the direct application of NTAPPs, open perspectives for the use of PAW in oral surgery and implantology. PAW could be used even more easily in these procedures, such as surgeries, for the removal of oral lesions and tooth extractions, especially in impacted third molars. Thus, the use of PAW in dentistry should be considered, given the antimicrobial, anti-inflammatory and wound healing properties of NTAPPs, in addition to their ability to alter the surface of dental implants. The simplicity of the technique, considering the use of PAW as a mouthwash or an irrigation agent and possibly the lower price of PAW compared to the direct application of NTAPPs, which would necessarily demand a device in the dental office, make PAW a potential adjuvant oral tool for conditions requiring tissue repair.

### 3.4. Anti-Cancer Therapy

Sensitivity of cancer cells to NTAPPs has been demonstrated in many studies. Reactive oxygen and nitrogen species may penetrate cancer cells more easily, compared to healthy ones, make them more vulnerable to their harmful effects. This fact may be explained by the higher amount of water channels (aquaporins) in cancer cells, that facilitates the transport of reactive species into cytosol. Additionally, the lipid peroxidation caused by free radicals generates pores in the membrane, which also allow the entry of reactive species into the cell. This process is attenuated by the condensation of membrane lipids in normal cells that are rich in cholesterol. However, cancer cells usually present fewer amount of lipids, which impairs this defense mechanism. The large influx of reactive species into cancer cells triggers signaling cascade pathways that may culminate in different types of cell death, such as apoptosis, necrosis or senescence, depending on the dose of exposure. Another important anti-cancer molecular mechanism of NTAPPs is their capacity to reduce the expression of some integrins. These molecules are essential for the adhesion, migration and invasion of cancer, which indicates that NTAPPs can be useful against metastases [38].

Besides the direct action of NTAPPs in cancer cells, they may be useful in this approach by their interaction with the tumor microenvironment. Reactive oxygen and nitrogen species are able to damage important extracellular matrix components, such as collagen, fibronectin and hyaluronic acid. The induction of an antitumor immunity has also been proposed as an action mechanism [38]. Anti-cancer properties of NTAPPs have been observed against many types of cancer cells [155,156,157,158,159]. Interestingly, clinical reports have already been conducted showing the role of NTAPPs in advanced squamous cell carcinoma (SCC) [153,160,161], most of them in intraoral sites [153,160]. SCC is the most prevalent oral cancer. More than 90% of the cases occur in men over 45 years of age, exposed to tobacco and/or alcohol. The lip is the most prevalent site, followed by the tongue [162]. An improvement in the quality of life of patients with advanced SCC located at intra-oral sites or the jaw was described, after NTAPPs treatment, by the reduction in odor and pain. Partial remission of the lesion occurred in some cases [153,160]. Additionally, a reduction in microbial load, wound healing of some infected ulcerated areas [153] and enhancement of apoptotic cells were described [161]. Partial or total remission of pre-malignant skin lesions, resulting from chronic ultraviolet exposure, referred to as actinic keratoses, were also observed after NTAPP treatment [163], which opens perspectives for actinic cheilitis, the pre-malignant lip counterpart, that precedes the emergence of lip SCC [164]. The adjunct treatment of initial SCC with NTAPPs has not been evaluated yet and it would be interesting, considering some in vitro responses of oral SCC to NTAPPs. A synergistic effect of cisplatin and NTAPP against oral SCC cells in vitro was described, associated with low cytotoxicity to normal oral cells [165]. Moreover, a combination of NTAPP with cetuximab inhibited invasion/migration of cetuximab-resistant oral SCC cells in vitro [166].

The possibility of using this technology of plasma-activated liquid is promising considering that PAL/PAW may be injected into large or deep tumors, facilitating the action in the entire lesion. Moreover, this kind of treatment would probably be faster and easier for the clinician compared to NTAPPs and more comfortable for the patient who is usually weakened by radiotherapy and/or chemotherapy. Treatment using liquids activated directly on the substrate or indirectly (activated first and then in contact with the substrate later) has been performed satisfactorily in many types of cancer cells with the use of different liquids, such as deionized water, cell culture media, Ringer’s solution and saline [17,18,167,168,169,170,171]. Apoptotic cells were observed in cancer cells exposed to activated deionized water [18,171]. Different studies have shown that PAL is also efficient against cancer, due to the toxic effects of oxygen and nitrogen species that are accumulated in these solutions and also by the immuno-stimulatory properties [172]. Different cancer cells may respond to PAL with decreased proliferation and migration and increased cell death by apoptosis, necrosis, autophagy and senescence [173]. A reduction in tumor burden and a metastasis-inhibitory effect were also observed with the use of PAL [169]. It was suggested that RNS could play a more relevant role in cancer cell death than ROS [171].

A previous study evaluating the effect of PAL on an oral squamous cell line (SCC15), observed an anti-cancer capacity of the plasma-activated medium. A reduction in cell viability was observed with an increasing incubation time. Moreover, they have demonstrated that many signaling pathways, such as p-53 pathway, could play a critical role in this process [174]. The effectiveness of PAW, as well as its possible mechanisms of action in oral cancer, has not been evaluated yet. Considering that the treatment of SCC is the entire removal of the lesion, this kind of treatment could be useful in two situations: (1) Prior to surgery, by washing the lesion or through the injection of plasma-activated water in deep neoplasms and (2) After the surgery, by reducing the microbial load and favoring wound healing. The anti-cancer properties could also be positive to avoid recurrences. The role of plasma-activated water in oral premalignant lesions, such as oral leukoplakia, erythroplasia and actinic cheilitis, should also be evaluated. Different kinds of treatments have been used in patients with actinic cheilitis, such as combinatory treatment with PDT and laser ablation. However, carbon dioxide laser ablation and vermilionectomy, that are invasive for the patients, have been considered the most effective treatments [164]. In this way, plasma-activated water could represent a non-invasive approach to be used in oral premalignant lesions with other therapies or even alone, depending on the results and risk factors of each patient.

### 3.5. Tooth Bleaching

Some studies showed that NTAPPs may be efficient for tooth bleaching, with a synergistic effect with other whitening agents [175,176]. In addition to lower concentrations of hydrogen peroxide solution (HP), its applicability might replace conventional light sources that present some limitations, such as questionable efficacy and high temperatures [176]. The association of HP and plasma exposure for 10 min generated a three-times higher bleaching than only HP, which probably occurred due to the presence of •OH, that was mostly present in the plasma-treated groups [175]. In addition to its efficiency for tooth bleaching, it was demonstrated that NTAPPs do not promote thermal damage or inflammatory responses in the pulp and oral soft tissues [177].

The role of NTAPPs applied to liquids for tooth bleaching has already been analyzed with interesting results. A helium-based NTAPP applied to the tooth surface with saline was evaluated. According to the authors, the wettability would enable the reactivity of ROS in the solution, attenuating the tooth dye, and it also would reduce the amounts of toxic gas produced using the air plasma method. The bleaching efficacy after 20 min of treatment was improved. It was 2.4 times higher than the effect produced by the whitening agent (35% of HP). The authors observed that H_2_O_2_ and •OH were generated in the saline solution, probably being a key factor for the observed effectiveness. Moreover, a scanning electron microscope (SEM) indicated smoother surfaces in the group treated with NTAPP + saline, which was probably less harmful to the enamel [178].

Satisfactory results of NTAPPs for tooth bleaching have also been observed in the presence of water. A previous study demonstrated that deionized water activated by NTAPP for 5 or 10 min, on the surface of the teeth, showed similar whitening presented by HP, with similar color stability [179]. Another piece of work that evaluated the whitening properties of bleaching agents and deionized water activated by NTAPP, in the pulp chamber of non-vital teeth, also obtained interesting findings. A total of 50 µL of water or bleaching agents were put in the pulp chamber, followed by plasma discharge, for 5 min. Interestingly, in addition to the improved bleaching of the whitening agents promoted by NTAPP, the generation of PAW was also effective. These findings indicate that PAW generated in the pulp chamber could be used as a substitute for conventional tooth bleaching in cases of non-vital teeth [180]. Considering the available results, PAW or saline generated on the surface of the teeth seems to be effective for tooth whitening, by the formation of H_2_O_2_ and •OH. Further studies are still required to investigate the effectiveness and to rule out possible toxicity effects of the reactive species to vital teeth. Moreover, studies evaluating the tooth bleaching potential of prior activated water (indirect method) or other plasma-activated liquids are also needed, since its application would be easier and possibly performed at home.

## 4. Conclusions

Plasma-activated water demonstrates antimicrobial activity, with promising applicability in both the decontamination of dental devices and the treatment of oral infectious diseases. The anti-inflammatory properties and wound healing benefits of PAW suggest that, in addition to its antimicrobial effect, PAW could favor the repair of previously infected lesions. In vivo studies are still needed to prove this effectiveness in oral diseases and rule out damage to the host. While RONS generated in PAW seems to present a fundamental role in decontamination and wound healing, the specific constituents H_2_O_2_ and •OH generated in plasma-activated liquids on the tooth surface may favor tooth bleaching. The findings related to plasma-activated water and liquids indicate that they could play an important role in the adjuvant treatment of some cancers by their antitumor response. Studies of PAW involving oral cancer would be interesting to investigate its application in oral neoplasms and the exact mechanisms of action inherent to this effect.

## Figures and Tables

**Figure 1 ijms-23-04131-f001:**
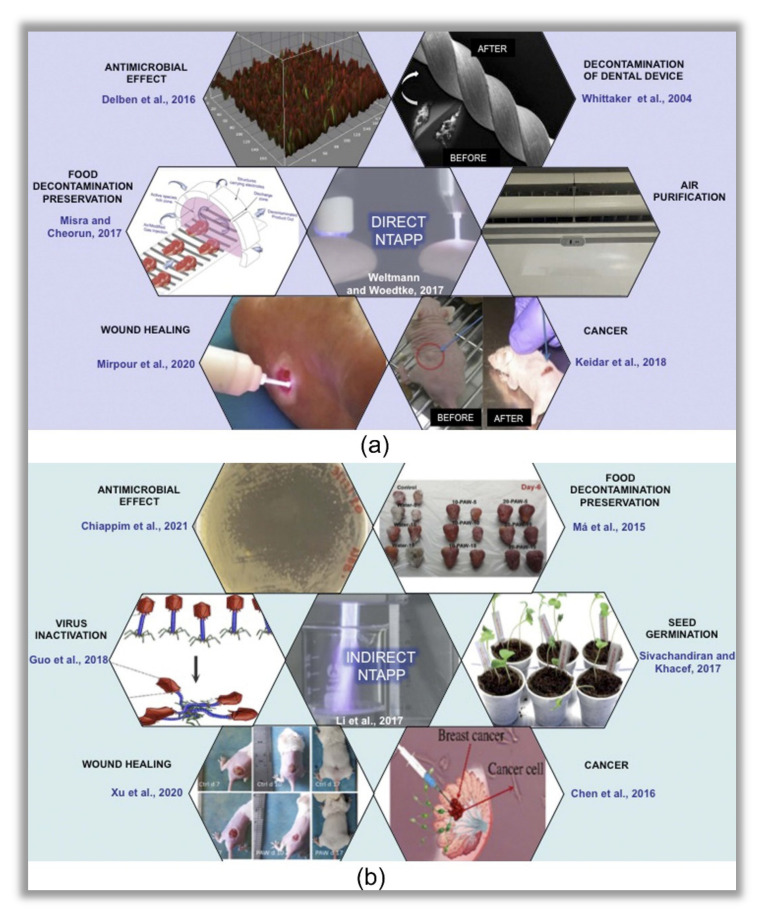
Schematic illustration of direct (**a**) and indirect (**b**) applications of non-thermal atmospheric pressure plasmas (NTAPPs). Note that due its antimicrobial effect, direct NTAPPs have been used for different purposes such as decontamination/preservation of food, air purification and decontamination of medical and dental devices. Additionally, an improvement in wound healing and anti-cancer properties has also been observed after direct exposure to NTAPPs. Similarly, the indirect modality has demonstrated antimicrobial effect against some types of microorganisms, with applicability in different fields. The reactive oxygen and nitrogen species generated in the liquids after plasma exposure may favor the seed germination. In addition, good responses have also been observed in wound healing and cancer treatment, which brings good perspectives for healthcare due to the clinical advantages of plasma-activated water (PAW) compared to direct NTAPPs. Some figures were reprinted with permission from Delben et al. [25], CC BY 4.0 license (2016); Whittaker et al. [26], copyright Elsevier (2004); Weltmann and Woedtke, [7], copyright IOP (2017); Misra and Cheorun, [27], copyright elsevier (2017); Keidar et al. [28], CC BY 4.0 license (2011); Sivachandiran and Khacef, [29], CC BY 3.0 license (2017); Guo et al. [15], CC BY 4.0 license (2018); Chen et al. [18], copyright Wiley (2016); Xu et al. [19], CC BY 4.0 license (2020); Li et al. [22], copyright Wiley (2017); Ma et al. [30], copyright Elsevier (2015); Chiappim et al. [31], copyright Wiley (2021).

**Figure 2 ijms-23-04131-f002:**
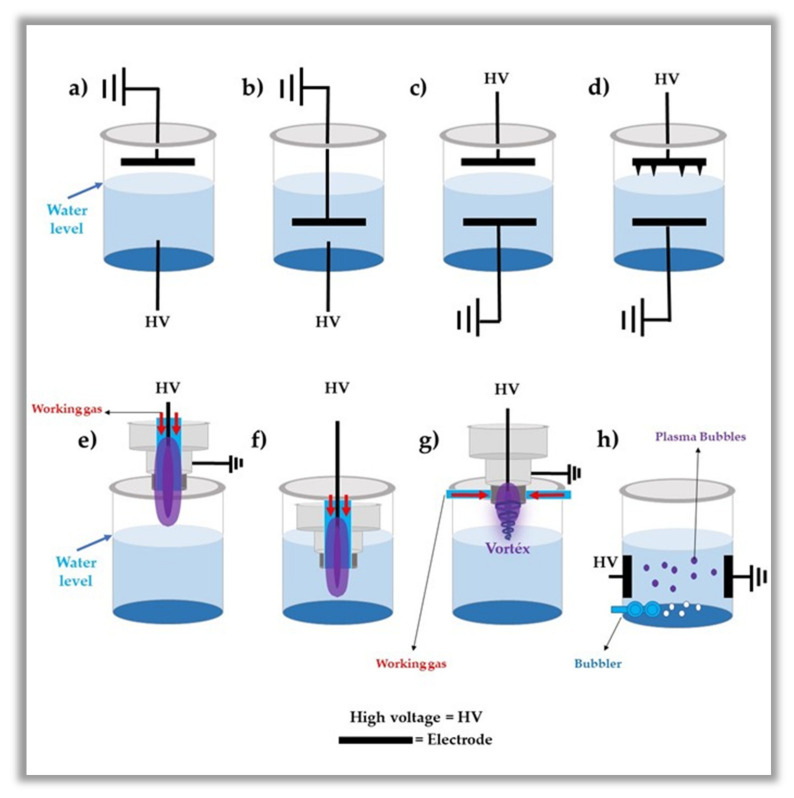
Schematic drawing of different discharges used for the preparation of plasma-activated water [47,74] (**a**–**c**) direct discharge into the water and (**d**) direct discharge into the water with multi-electrodes; (**e**) discharge in the gaseous phase onto the water surface; (**f**) discharge in the gaseous phase into the water; (**g**) discharge in the gas phase onto the water surface with plasma generated on forward vortex flow reactor (FVFR); (**h**) discharges in bubbles into the water.

**Figure 3 ijms-23-04131-f003:**
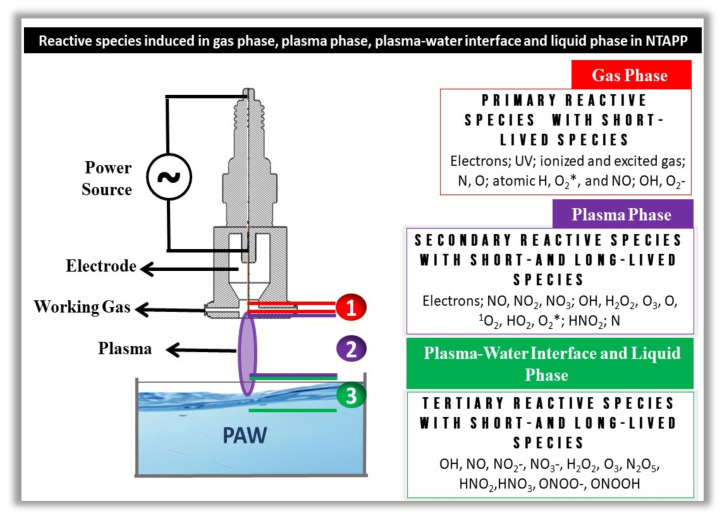
Schematic diagram that shows the regions of reactive species generation induced by NTAPP used to activate water. Note the three regions of RONS formation in an NTAPP device in contact with water. The primary reactive species, short-lived species, electrons and electromagnetic radiation in the ultraviolet range are found in region 1 and are in the gas phase. In region 2, we can observe the secondary reactive species made up of short-lived and long-lived species, and this region is contained in the plasma phase. Tertiary reactive species are short-lived and long-lived species, and their region is limited to the plasma–water interface and liquid phase.

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
