# Peer review of "Applications of Plasma-Activated Water in Dentistry: A Review"

_ijms, 2022, doi:10.3390/ijms23084131_

Round 1
Reviewer 1 Report
“Plasma dentistry” is an entirely new and fast emerging field of non-temperature plasmas (NTP) applications, which attracts a lot of researcher’s attention during the last decade. By now, there are a large number of papers describing beneficial effects of non-temperature plasmas for dentistry and their amount continues to grow. Perspectives of the NTPs indirect usage (primarily water activated by non-temperature atmospheric plasma) in clinical practice are a focus of resent discussions. Thus, the topic of the present manuscript is of current interest for both the plasma research community and practicing dentists.
The title accurately describes the main subject of the review. The authors reviewed a sufficient number of articles published from the beginning of the 2000s up to nowadays. The material have been analyzed also in the context of the interactions of the NTP-produced active species with living cells and oral tissues. I find the content to appropriate for the International Journal of Molecular Sciences.
However, the manuscript needs further improvements before the acceptance. My comments are as follows:
- The authors use two abbreviations namely NTAPP and CAP for non-thermal atmospheric pressure plasma and cold atmospheric plasma respectively. I believe that NTAPP and CAP mean the same plasmas type and it is better to use only one abbreviation throughout the text to avoid possible misunderstanding. Otherwise, the difference between NTAPP and CAP should be clearly specified in the Introduction.
- In the Introduction the authors mention that plasma-activated water and liquids are mainly applied for medical purposes (lines 57-59). However, various application of plasma-activated water in agriculture are intensively studied and there are a lot of papers (original and reviews) devoted to this problem. In addition, agricultural applications of plasma-activated water are included in the Figure 1. Thus, I believe the given statement should be improved.
- Lines 77-80: Changes are required. Various types of radiation are generated as result of plasma chemical reactions, so “consist” is not really an appropriate term in this case. Term “reactive species” should be explained in detail, because ions and metastable species are also responsible for some plasma-stimulated processes. In addition, may be it is better to use “particles” instead of “species”.
- Lines 98-105: There are only four sections in the Review and Section 4 is Conclusions. Where is Section 5?
- Section 2 Plasma-activated and Plasma-treated water should be improved. Is not plasma-treated water activated due to plasma exposure? Are amounts of liquid to be treated the only difference between plasma-activated and Plasma-treated water? How much is it - small amounts? Line 110: “with the aid of a reactive” – does it mean “activator”? Lines 112-113: identical words with “active”.
- Section 2.1: Main Reactors of Producing: Additional references concerning the design and characteristics of plasma generators (e.g. frequency, power, etc.) which were used for water treatment as well as information about treatment conditions should be included into the text. Several images of such plasma generators would be nice also. Do any commercial reactors exist? Please give brief speculations whether existing plasma devices are applicable and convenient for clinical practice.
- Subsections 2.1.1 and 2.2.2: Taking into account the subject area of IJMS and in order to make the text easier to the readers it is better to present the given reactions as a separate scheme or to combine them with the Figure 2. The values of the ionization energies and the rate constants can be also excluded, because such data are essential mainly for plasma physics and chemistry papers.
- Line 147: “the dissolution of the NOx”. Does it mean “the NOx conversion”?
- Line 156: “the 155 pH of the water also decreases, which can be seen by reactions (8) and (11)”. May be (9) instead of (11)? Please, check.
- Lines 162-163: “also observed, as can be 162 observed”. Should be improved.
- The numbering of Section 2 seems to be incorrect. It is worth placing Section 2.2 before Subsection 2.1.1, as it looks like the introduction to the plasma chemical part. On the other hand, Section 2.2 contains Subsections 2.2.2 and 2.2.3 (Hydrogen peroxide and Ozone), though the Section 2.2 is “The Origin of Reactive Oxygen and Nitrogen Species”. Please, check.
- Line 224: the radical symbol is missing.
- Line 290: “As discussed in the last section” should be clarified.
- Line 294: “plasma discharge in both conditions”. Please, comment what are these conditions.
- Line 313: Please give the description of the abbreviation DUWLs here.
- Lines 334-339; 370-374; 386-390; 470-491; 530-548; 619-636: The main subject of the present Review is the applications of plasma-activated water in dentistry. Nevertheless, several parts of the paper consider the effects occurring in non-oral cells and tissues (e.g. skin, ovarian), and system which are not used in dentistry (e.g. duonenoscope). These parts should be shortened to add more information about plasma dental applications instead.
- Lines 379, 391, 600: references numbers are missing.
- Line 460: Please, comment what “good effect” are.
- Line 656: “the presence of OH-“. Does it mean OH radical? Please, check.
- Line 668: Please give the description of the abbreviation SEM here.
Author Response
We appreciate the time and effort that the reviewers have dedicated to providing their valuable feedback on our manuscript. We are grateful to the reviewers for their insightful comments on our paper. We have been able to incorporate changes to reflect most of the suggestions provided by the reviewers. We have highlighted the revisions within the manuscript.
Below is a point-by-point response to the reviewers’ comments and concerns.
Reviewer Comments, Author Responses, and Manuscript Changes
REVIEWER 1
Comment 1:
1) The authors use two abbreviations namely NTAPP and CAP for non-thermal atmospheric pressure plasma and cold atmospheric plasma respectively. I believe that NTAPP and CAP mean the same plasmas type and it is better to use only one abbreviation throughout the text to avoid possible misunderstanding. Otherwise, the difference between NTAPP and CAP should be clearly specified in the Introduction.
Response 1:Thanks for pointing this out. To avoid possible misunderstandings, we opted for the abbreviation NTAPP, and throughout the manuscript, we replaced the abbreviation CAP with NTAPP. All substitutions are highlighted in yellow.
Comment 2:
2) In the Introduction the authors mention that plasma-activated water and liquids are mainly applied for medical purposes (lines 57-59). However, various application of plasma-activated water in agriculture are intensively studied and there are a lot of papers (original and reviews) devoted to this problem. In addition, agricultural applications of plasma-activated water are included in the Figure 1. Thus, I believe the given statement should be improved.
Response 2:Thanks for pointing this out. To improve the introduction and add the due relevance of PAW in agriculture and other application areas, we have added the following paragraph between lines 50 and 61.
“To exemplify, Table 1shows the main review articles published in the last 3 years on applications of direct and indirect NTAPPs, in areas ranging from medicine, biomedicine, dentistry, agriculture, and the food industry. It is worth mentioning that although the present work is focused on dentistry, the direct and indirect NTAPPs are of great importance for agriculture and the food industry. This wide range of applications of direct and indirect NTAPPs is schematically presented in Figure 1. Figure 1a highlights the applicability of direct NTAPPs that cover different human and food health care areas. Figure 1b illustrates the applications of NTAPPs through the modality generating plasma-activated liquid (PAL) or PAW, which are later applied to substrates. As described, the PAW and PAL studies are focused on different purposes, which shows the effort of many scientists to improve and make plasma technology more accessible.”
Additionally, we add the following paragraph between lines 80 and 88. Table 1 was also added with the main review articles published in the last 3 years. In this way, we believe that it was possible to improve the introduction as suggested by the reviewer.
“Due to the growing interest of the scientific community in plasma technology applied to dentistry, a review article is presented here to serve as a quick guide for dentists, physicians, physicists, engineers, and health professionals in general. This review aims to discuss the applicability of PAW in dentistry, which, as can be seen in Table 1, is a review with little or no exploration. Thus, based on the published literature, the findings of PAW applied to dentistry are presented, as well as the future perspectives for the area considering the main results related to direct and indirect NTAPP for biomedical applications.
Table 1. Summary of the main review articles published in the last 3 years on applications of direct and indirect NTAPPs.
|
Approach |
NTAPP modalities |
Publication year/ Reference |
|
|
Molecular Mechanisms of the Efficacy of NTAPP in Cancer Treatment |
Direct |
2020 [X] |
|
|
A Powerful Tool for Modern Medicine |
Direct |
2020 [X] |
|
|
The New Frontier in Low Temperature Plasma Applications |
Direct |
2020 [X] |
|
|
Atmospheric Cold Plasma Treatment in Fruit Juices |
Direct |
2020 [X] |
|
|
Chemical, Physical and Physiological Quality Attributes of Fruit and Vegetables Induced by Cold Plasma Treatment |
Direct |
2020 [X] |
|
|
Cold Plasma as a New Hope in the Field of Virus Inactivation |
Direct |
2020 [X] |
|
|
The Effects of Plasma on Plant Growth, Development, and Sustainability |
Direct |
2020 [X] |
|
|
Cold Plasma for the Control of Biofilms in Food Industry |
Direct |
2020 [X] |
|
|
Potential of Cold Plasma Technology in Ensuring the Safety of Foods and Agricultural Produce |
Direct |
2020 [X] |
|
|
Plasma Agriculture from Laboratory to Farm |
Direct |
2020 [X] |
|
|
Aurora Borealis in Dentistry |
Direct |
2021 [X] |
|
|
Applications of Cold Atmospheric Pressure Plasma in Dentistry |
Direct |
2021 [X] |
|
|
Cold Atmospheric Pressure Plasma Technology in Medicine, Agriculture and Food Industry |
Direct |
2021 [X] |
|
|
The Antimicrobial Effect of Cold Atmospheric Plasma against Dental Pathogens |
Direct |
2021 [X] |
|
|
Influence of Atmospheric Cold Plasma on Spore Inactivation |
Direct |
2021 [X] |
|
|
Plasma-Assisted Agriculture: History, Presence, and Prospects |
Direct |
2021 [X] |
|
|
Improving Seed Germination by Cold Atmospheric Plasma |
Direct |
2022 [X] |
|
|
Reactive Nitrogen Species in Plasma-Activated Water |
Indirect |
2020 [X] |
|
|
Recent Advances in Plasma-Activated Water for Food Safety |
Indirect |
2022 [X] |
|
|
Influence of Plasma-Activated Water on Physical and Physical–Chemical Soil Properties |
Indirect |
2020 [X] |
|
|
PAW Triggers Plant Defense Responses |
Indirect |
2020 [X] |
|
|
PAW Generation, Origin of Reactive Species and Biological Applications |
Indirect |
2020 [X] |
|
|
Interactions of Plasma-Activated Water with Biofilms |
Indirect |
2020 [X] |
|
|
A Comprehensive Review of PAW for Enhanced Food Safety and Quality |
Indirect |
2021 [X] |
|
|
Applications of PAW in the Food Industry |
Indirect |
2020 [X] |
|
|
PAW for Cancer Treatment: Positives, Potentials and Problems of Clinical Translation |
Indirect |
2020 [X] |
|
|
Review on Discharge Plasma for Water Treatment |
Indirect |
2020 [X] |
|
|
PAW on Microbial Growth and Storage Quality of Fresh-cut Apple |
Indirect |
2020 [X] |
|
|
PAW Production and its Application in Agriculture |
Indirect |
2021 [X] |
|
|
Diagnostic Analysis of Reactive Species in PAW: Current Advances and Outlooks |
Indirect |
2021 [X] |
|
|
PAW from DBD as a source of Nitrogen for Agriculture |
Indirect |
2021 [X] |
|
|
PAW, Hydrogen Peroxide, and Nitrates on Lettuce Growth |
Indirect |
2021 [X] |
|
All the new references of the the table and main text have been added to the manuscript.
Comment 3:
3) Lines 77-80: Changes are required. Various types of radiation are generated as result of plasma chemical reactions, so “consist” is not really an appropriate term in this case. Term “reactive species” should be explained in detail, because ions and metastable species are also responsible for some plasma-stimulated processes. In addition, may be it is better to use “particles” instead of “species”.
Response 3:Thanks for pointing this out. To clarify this point, we added the following paragraph between lines 62 and 68.
“To understand the mechanisms of action related to NTAPPs, it is essential to know that the plasma is a gas partially ionized consisting of particles as electrons, ions, metastable species, radiation as ultraviolet (UV), visible (VIS), electromagnetic field, and reactive species. It is worth highlighting that the term reactive species is commonly used for free radicals and reactive oxygen species (ROS), and reactive nitrogen species (RNS). It is highly reactive is basically due to the presence of unpaired valence electrons or non-static bonds on its structure.”
Comment 4:
4) Lines 98-105: There are only four sections in the Review and Section 4 is Conclusions. Where is Section 5?
Response 4:Thanks for pointing this out. We fixed this error.
Comment 5:
5) Section 2 Plasma-activated and Plasma-treated water should be improved. Is not plasma-treated water activated due to plasma exposure? Are amounts of liquid to be treated the only difference between plasma-activated and Plasma-treated water? How much is it - small amounts? Line 110: “with the aid of a reactive” – does it mean “activator”? Lines 112-113: identical words with “active”.
Response 5:Thanks for pointing this out. We have rewritten section 2 to improve understanding and to avoid confusion. To clarify this point, we added the following paragraph between lines 116 and 128.
“In the literature, the terms treatment and activation are often misused. To avoid doubt, treatment is defined as practical means or refinements used to combat/mitigate a problem. In the specific case of water treated by plasma, the term treated refers to the process of elimination or complete mineralization in wastewater of synthetic dyes, pharmaceuticals products, pathogenic bacteria, among other pollutants. Therefore, plasma water treatment is commonly used to purify or decontaminate small or large amounts of water. In contrast, activation is understood as increased activity, becoming active, boosting, accelerating, or intensifying some specific property. In this case, the non-pollutant water exposed to the plasma becomes activated, i.e., it obtains new properties. Unlike treatment, activation is carried out in small amounts of water (between 1 mL to 500 mL), but the reactors used are the same with a slight modification in both cases. It is important to note that deionized water, distilled water, filtered water, and potable tap water, i.e., pure water without pollutants, are usually used for activation.”
Comment 6:
6) Section 2.1: Main Reactors of Producing: Additional references concerning the design and characteristics of plasma generators (e.g. frequency, power, etc.) which were used for water treatment as well as information about treatment conditions should be included into the text. Several images of such plasma generators would be nice also. Do any commercial reactors exist? Please give brief speculations whether existing plasma devices are applicable and convenient for clinical practice.
Response 6:Thanks for pointing this out. We have rewritten section 2.1 to improve understanding and to avoid confusion. To clarify these points, we added Figure 2 and the following paragraph between lines 148 and 181.
“It is important to note that NTAPPs operate at high voltage. As reported in the literature, these voltages range from 1 to 50 kV, with operating frequencies that can start in tens to thousands of Hz (kHz) and powers that generally do not exceed values greater than 10 W. Commonly used working gases are helium (He), argon (Ar), oxygen (O2), nitrogen (N2), air, or a mixture of these gases. Working gases are used with flow rates ranging from 1 to 30 L/min [X-X]. Indeed, there is a range of reactors used for water and liquid activation, and every day a new article is published with new reactors that have minor changes. Therefore, a dedicated review article would be needed to demonstrate the NTAPP generation reactors and their characteristics, but the focus of the present work is not that. Thus, below we show some reactors used to generate PAW.
(See figure 2 in the manuscript)
“As seen in Figure 2, discharges are used directly into the water and on the surface of the water, which considerably affects the chemical composition of the PAW. This difference in chemical composition is basically due to the differences between the rupture forces in the gas phase (discharge on the surface) and the water (discharge inside) [X, X]. However, as reported in the literature, the most applied systems are those that operate with plasma discharge in contact with water, i.e., PJ and DBD as a plasma source [X, X]. As demonstrated in the next section, these plasmas deliver the RONS from the plasma gas to the liquid phase more efficiently. “
“An important question is whether there is commercial equipment that can be applied in clinical practice. Recently, Andrasch et al. [X], Pemen et al. [X38], and Schnabel et al. [X] developed pilot units with potential for practical applications. Andrasch et al. [X] and Schnabel et al. [X39] obtained a PAW production rate of 1L/min. Pemen et al. [X] obtained 0.5 L/min. However, these devices cannot meet the application requirements in dentistry and medicine, which are low pH and high concentrations of RONS. On the other hand, in agriculture and in the food industry, the production of millions of liters of PAW at a low cost is required. Therefore, it can be said that PAW generating equipment is still in the pilot phase and has great commercialization potential in the coming years.”
Comment 7:
7) Subsections 2.1.1 and 2.2.2: Taking into account the subject area of IJMS and in order to make the text easier to the readers it is better to present the given reactions as a separate scheme or to combine them with the Figure 2. The values of the ionization energies and the rate constants can be also excluded, because such data are essential mainly for plasma physics and chemistry papers.
Response 7:Thanks for pointing that out. We appreciate your comments and have removed the ionization energy and rate constant values to make the manuscript easier to read. Subsections 2.1.1, 2.2.2. and 2.2.3. were extracted from the main text and added to the appendix.
Comment 8:
8) Line 147: “the dissolution of the NOx”. Does it mean “the NOx conversion”?
Response 8:Thanks for pointing this out. We fixed this error.
Comment 9:
9) Line 156: “the 155 pH of the water also decreases, which can be seen by reactions (8) and (11)”. May be (9) instead of (11)? Please, check.
Response 9:Thanks for pointing out this error. We fixed this error.
Comment 10:
10) Lines 162-163: “also observed, as can be 162 observed”. Should be improved.
Response 10:Thanks for pointing this out. We fixed this error and improved “also observed, as showed...”
Comment 11:
11) The numbering of Section 2 seems to be incorrect. It is worth placing Section 2.2 before Subsection 2.1.1, as it looks like the introduction to the plasma chemical part. On the other hand, Section 2.2 contains Subsections 2.2.2 and 2.2.3 (Hydrogen peroxide and Ozone), though the Section 2.2 is “The Origin of Reactive Oxygen and Nitrogen Species”. Please, check.
Response 11:Thanks for pointing this out. We have rewritten section 2.1 to improve understanding and avoid confusion, as seen in response 6. We have replaced the sections to better adjust the present work.
Comment 12:
12) Line 224: the radical symbol is missing.
Response 12:Thanks for pointing out this error. We fixed this error.
Comment 13:
13) Line 290: “As discussed in the last section” should be clarified.
Response 13:Thanks for pointing this out. To avoid repetition, we removed this statement because the details about reactive species were provided in the section 2.
Comment 14:
14) Line 294: “plasma discharge in both conditions”. Please, comment what are these conditions.
Response 14:Thanks for pointing this out. These conditions are the application of NTAPP over the water, inside the water, and touching the water surface. We fixed this error and replaced this statement in the first paragraph of the section 3.1 (Decontamination of dental devices), as follows:
“In addition to being useful in the inactivation of microorganisms when applied over, inside, or touching the water surface [X], NTAPPs are also efficient against prion proteins [X], which are resistant to traditional cleaning methods [X]”.
Comment 15:
15) Line 313: Please give the description of the abbreviation DUWLs here.
Response 15:Thanks for pointing this out. The description of the abbreviation DUWLs (unit waterline system tubes ) was inserted in the text in the section 3.1 (Decontamination of dental devices), as follows:
"The decontamination of dental unit waterline system tubes (DUWLs) is a challenge for dentistry due to the risk of cross-contamination [X]"
Comment 16:
16) Lines 334-339; 370-374; 386-390; 470-491; 530-548; 619-636: The main subject of the present Review is the applications of plasma-activated water in dentistry. Nevertheless, several parts of the paper consider the effects occurring in non-oral cells and tissues (e.g. skin, ovarian), and system which are not used in dentistry (e.g. duonenoscope). These parts should be shortened to add more information about plasma dental applications instead.
Response 16:Thanks for pointing this out. The effects of NTAPP and PAW/PAL in non-oral cells and tissues were shortened in the sections 3.3 (Anti-inflammatory properties and wound healing) and 3.4 (Anti-cancer therapy) in order to make the manuscript more focused on dentistry. The sterilization of the duodenoscope by PAW only was kept because this material was coated by polymer resin, that is a kind of material important for dentistry. We inserted the explanation in the text “Resin based materials have important applicability in dentistry [X]”.
A new reference about the effect of PAW intake on the mineral composition and surface features of mouse teeth and also its oral toxicity evaluation was inserted in the first paragraph of the section 3.3. (Anti-inflammatory properties and wound healing). We have also added a new reference about the effect of plasma activated-liquids (PAL) on an oral squamous cell line in the last paragraph of the section 3.4 (Anti-cancer therapy). Additionally, in the sections 3.2 (treatment of oral infections diseases) and 3.3 (Anti-inflammatory properties and wound healing) were inserted more references/informations about NTAPP in dentistry. The role of NF-κB e STAT3 signaling pathways in dentistry was also explained considering that PAW/ PAL may act in these pathways in some cells.
The following statements related to dentistry were inserted:
“Resin based materials have important applicability in dentistry [X]”
“In the context of cariology it is also important to emphasize the positive property of NTAPPs in adhesive restorations. Plasma exposure generates the deposition of free radicals and ions on tooth substrate changing the surface proteins of dentin what has led to increased bond strength in adhesive restorations. The enhance in bone bond strength avoid microleakage and consequently prevents secondary caries [X]. It is not known whether the PAW can also increase the bond strength improving the restoration performance. Probably, this process could also occur with plasma-liquid technology due to the action of the reactive species on the surface of dentin.”
“A previous study investigated the effect of PAW intake in mice after 90 days administration once its use in dentistry may lead to accidental ingestion. The mineral composition and surface micro-morphology of vital mouse teeth after long-term exposure, as well as local and sistemic toxicity were evaluated. The authors observed that there were not significant changes in the mineral composition and surface micro-morphology of the teeth. Moreover, the long-term exposure was not toxic to tongue, oral mucosa, sublingual glands or other body organs, which presented normal structure and physiology [X]. In addition to not being harmful to mammalian cells, NTAPPs have demonstrated to decrease inflammation and contribute to tissue repair [X].”
“These effects have also been observed in oral studies. The treatment of oral candidiasis in mice showed low occurrence of inflammatory alterations. After plasma exposure, the cell inflammatory infiltrate was predominantly mononuclear and macrophage-rich, with scarce polymorphonuclear cells [116]. Additionally, an study evaluating the role of NTAPP as an adjuvant therapy for the treatment of periodontitis induced in rats observed that the expression of inflammatory-related cytokines such as TNF-α and IL-1β decreased significantly in the group where NTAPP was used as an adjuvant approach, while the level of the anti-inflammatory cytokine IL-10 showed a significant increase [X].”
“In this way, plasma activated liquids have shown to act in different inflammatory signaling pathways of keratinocytes. These findings are important for dentistry once NF-κB e STAT3 pathways are involved in the etiopathogenesis of periodontitis [138]. Moreover, these pathways also play a role in oral candidiasis once mucosal candidiasis promotes NF-κB activation [X]and STAT3 signaling are related to IL-17-mediated immunity to oral mucosal candidiasis [X].”
“Cell proliferation and migration were observed in human keratinocytes exposed to 15 seconds of medium activated with Helium and Argon (He/Ar)-generated NTAPP [X], which could favor the re-epithelialization of wounds on the skin and also on the oral mucosa that has keratinocytes in the epithelial composition. There is no study evaluating the effect of PAW or PAL on mouth wound healing although two studies carried out in rats and mice have observed a tendency to improve periodontal tissue loss after the treatment of experimental peridontitis with NTAPPs [X,X]. Moreover, wound healing of some infected ulcerated areas of advanced oral squamous cell carcinoma was observed after NTAPP exposure [X]. Considering that wound repair is important for different modalities of dentistry such as oral surgery, periodontics, oral pathology and implantology, PAW could be an option to accelerate the healing after oral diseases or oral surgeries.”
“The oral implant modification with NTAPPs has also been suggested since this treatment may enhance the roughness and wettability of the implant surface thus improving the cell adhesion and consequently the osseointegration [X]. These results with direct application of NTAPPs open perspectives for the use of PAW in oral surgery and implantodology.”
“Thus, the use of PAW in destistry should be considered given the antimicrobial, anti-inflammatory and wound healing properties of NTAPPs, in addition to its ability to alter the surface of dental implants. ”
“A previous study evaluating the effect of PAL on an oral squamous cell line (SCC15) observed anti-cancer capacity of the plasma-activated medium. Reduction in cell viability was observed with increasing incubation time. Moreover, they have demonstrated that many signaling pathways, such as p-53 pathway, could play a critical role in this process [X].”
Comment 17:
17) Lines 379, 391, 600: references numbers are missing.
Response 17:Thanks for pointing this out. We inserted these references that were missing.
Comment 18:
18) Line 460: Please, comment what “good effect” are.
Response 18:Thanks for pointing this out. Actually, the good effects are the anti-inflammatory effect and the contribution to tissue repair. We changed the statement, as follows:
“In addition to not being harmful to mammalian cells, NTAPPs have demonstrated to decrease inflammation and contribute to tissue repair [X]”
Comment 19:
19) Line 656: “the presence of OH-“. Does it mean OH radical? Please, check.
Response 19:Thanks for pointing this out. Yes. It means ·OH. We fixed this error in the text in the sections 3.5 (Tooth bleaching) and 4 (conclusion).
Comment 20:
20) Line 668: Please give the description of the abbreviation SEM here.
Response 20:Thanks for pointing this out. We inserted the description of the abbreviation SEM (scanning electron microscope) in the section 3.5 (Tooth bleaching), as follows:
"Moreover, scanning electron microscope (SEM) indicated smoother surfaces in the group treated with NTAPP + saline, which probably was less harmful to enamel [X]."
Reviewer 2 Report
This review pretends reviewing and discuss the applications of Plasma-Activated Water in Dentistry. However, the review needs to be improved and better focused on dentistry applications to bring originality with respect to already existing reviews in the field of cold atmospheric plasmas for biomedical applications. Selection of the references as to be more precise regarding what the reader expects from this kind of review. Coordination in the writing and interconnection of the different parts as also to be enhanced as well as a synthetic effort must be done regarding the introduction & part 2 of the paper. At this time, I do not recommend its publication and I suggest to the authors the following changes/enhancements to improve the manuscript:
With such a title, the reader expects a review more focused on dentistry applications. The review needs to be more focused on. Starting with the title, “Applications of Plasma-Activated Water in Dentistry: a Review”, maybe the authors should consider to replace the word “water” by “liquid” since several indirect applications of APPs in dentistry contemplate physiological/saline solutions for plasma treatment.
A) Building of Part 2 is very confused and different concepts are mixed there.
1º/ It would be accurate to synthesized and combined this part with the introduction (it also should help for the focusing of the review). Part 2.1 “Main reactor of producing” with a unique subsection focused on “2.1.1. Nitrite, Nitrate, and Nitrous Acid” and part 2.2. “The Origin of Reactive Oxygen and Nitrogen Species” with subsections on “2.2.2. Hydrogen Peroxide” and “2.2.3. Ozone” do not have any logical conductive thread. It would be more accurate to move 2.1.1. into the part of “Nitrite, Nitrate, and Nitrous Acid” into the part 2.2. “The Origin of Reactive Oxygen and Nitrogen Species”. Anyway, in general, this part 2 on “Plasma-activated and plasma-treated water” is already tackled in many previous reviews, and some parts seems to be directly extracted/adapted from another ones, with missing references (Examples: great part of “2.1.1. or 2.2.2. can be found in previous review [33] without being mentioned as reference in this concrete parts. Interactions of generated RONS into liquid can be found in another uncited review of Buxton et al., Critical review of rate constants for reactions of hydrated electrons, hydrogen atoms and hydroxyl radicals (.OH/.O-) in aqueous solution. Journal of Physical and Chemical Reference Data, 1988, 17: 513). With the current number of previous reviews/publications focused on RONS generation in liquid by NTAPPs, the authors should consider removing the chemical equations from line 135 to 259, and make a sum-up (with the accurate references) of the different reactive species generated by NTAPPs in liquids without entering in so much details. It should take a couple of paragraphs and it would help for the clarity of the review.
2º/ Regarding the starting of the part 2 (L. 107) regarding what the authors defined as a current “misused” of “the terms treatment and activation”. The authors should consider limiting the writing as a definition of the different terms (This point is a minor detail/ personal opinion).
B) Graphical design of Figure 2 needs to be improved
C) Extensiveness of the references of the manuscript is not due to the extensiveness of the published works of plasma- treated liquid in the dentistry field but of a review not focused enough on the field. As reader, I expect references more focused on dentistry applications of NTAPPs/CAPs.
D) Introduced abbreviations in “Introduction” or Part 2. are once again defined in Part 3 (eg: CAP, 33 & 270). Moreover, it would be more accurate to include in the introduction the entire paragraph (l. 270) into the subsection “3. PAW Applied in Dentistry - 3.1. Decontamination of dental devices” regarding the introduction/definition of Cold Atmospheric Plasmas: “Cold atmospheric plasma (CAP) is a safe technology that uses low temperature without the generation of toxic residues and chemical products [68]. CAP present a harmful effect to different microbial species [72–74]. Charged particles, emission of UV radiation and specially the reactive oxygen and nitrogen species, generated by CAP, are considered the contributing factors for its antimicrobial property [75]. In addition to being useful in microbial decontamination process”.
Including some aspects are repeated from the introduction (l.77): “NTAPPs, it is important to know that it is an ionized gas consisting of electrons, ions, metastable species, ultraviolet (UV), visible (VIS) radiation, electromagnetic field, and reactive species”.
These details, among others, revealed a lack of coordination/interconnection in the writing of the different parts pf the manuscript.
(Personal opinion: The authors should also consider introducing/defining NTAPPs and CAPs at the beginning of the paper and use a unique abbreviation along the manuscript)
E) The authors should consider reducing the 2-pages section on “3.4. Anticancer therapies” (L 570) into the section “PAW Applied in Dentistry” and focusing just on oral cancers, especially on the neck squamous cell carcinoma (SCC), the main one (as correctly stated). However, this section includes extensive parts regarding the use of CAPs/NTAPPs and its proved effectiveness onto a broad range of kinds of cancers, that could be reduced to a simple paragraph with the adequate references.
F) *** Details: “RONs” -> “RONS” (l. 186)
*** In “3.4. Anticancer therapies”: “promising source (l. 584), … promising responses (l. 589) promising results (l. 592), “…is promising considering that”… (l. 610)”: Try to not overused the term “promising” and state/detail the important results of the cited reference regarding the promising aspects.
Author Response
RESPONSE TO REFEREES: IJMS - 1647046 entitled " Applications of Plasma-Activated Water in Dentistry: A Review"
We appreciate the time and effort that the reviewers have dedicated to providing their valuable feedback on our manuscript. We are grateful to the reviewers for their insightful comments on our paper. We have been able to incorporate changes to reflect most of the suggestions provided by the reviewers. We have highlighted the revisions within the manuscript.
Below is a point-by-point response to the reviewers’ comments and concerns.
REVIEWER 2
This review pretends reviewing and discuss the applications of Plasma-Activated Water in Dentistry. However, the review needs to be improved and better focused on dentistry applications to bring originality with respect to already existing reviews in the field of cold atmospheric plasmas for biomedical applications. Selection of the references as to be more precise regarding what the reader expects from this kind of review. Coordination in the writing and interconnection of the different parts as also to be enhanced as well as a synthetic effort must be done regarding the introduction & part 2 of the paper. At this time, I do not recommend its publication and I suggest to the authors the following changes/enhancements to improve the manuscript:
With such a title, the reader expects a review more focused on dentistry applications. The review needs to be more focused on. Starting with the title, “Applications of Plasma-Activated Water in Dentistry: a Review”, maybe the authors should consider to replace the word “water” by “liquid” since several indirect applications of APPs in dentistry contemplate physiological/saline solutions for plasma treatment.
- A) Building of Part 2 is very confused and different concepts are mixed there.
1º/ It would be accurate to synthesized and combined this part with the introduction (it also should help for the focusing of the review). Part 2.1 “Main reactor of producing” with a unique subsection focused on “2.1.1. Nitrite, Nitrate, and Nitrous Acid” and part 2.2. “The Origin of Reactive Oxygen and Nitrogen Species” with subsections on “2.2.2. Hydrogen Peroxide” and “2.2.3. Ozone” do not have any logical conductive thread. It would be more accurate to move 2.1.1. into the part of “Nitrite, Nitrate, and Nitrous Acid” into the part 2.2. “The Origin of Reactive Oxygen and Nitrogen Species”. Anyway, in general, this part 2 on “Plasma-activated and plasma-treated water” is already tackled in many previous reviews, and some parts seems to be directly extracted/adapted from another ones, with missing references (Examples: great part of “2.1.1. or 2.2.2. can be found in previous review [33] without being mentioned as reference in this concrete parts. Interactions of generated RONS into liquid can be found in another uncited review of Buxton et al., Critical review of rate constants for reactions of hydrated electrons, hydrogen atoms and hydroxyl radicals (.OH/.O-) in aqueous solution. Journal of Physical and Chemical Reference Data, 1988, 17: 513). With the current number of previous reviews/publications focused on RONS generation in liquid by NTAPPs, the authors should consider removing the chemical equations from line 135 to 259, and make a sum-up (with the accurate references) of the different reactive species generated by NTAPPs in liquids without entering in so much details. It should take a couple of paragraphs and it would help for the clarity of the review.
2º/ Regarding the starting of the part 2 (L. 107) regarding what the authors defined as a current “misused” of “the terms treatment and activation”. The authors should consider limiting the writing as a definition of the different terms (This point is a minor detail/ personal opinion).
Response :Thanks for pointing this out. We reworded the Introduction and Section 2 to improve understanding and avoid confusion. To clarify these points, we added Figure 2 and improved Figure 3. We added paragraphs: i) between lines 50 and 68; ii) between lines 80 and 88; iii) between lines 116 and 128, and iv) between lines 148 and 181. The subsections 2.1.1, 2.2.2. and 2.2.3. were extracted from the main text and added to the appendix. Additionally, we added the references Khlyustova et al. 2019 and Buxton et al. 1988 in the appendix. All these additions are highlighted in yellow in the text.
- B) Graphical design of Figure 2 needs to be improved
Response:Thanks for pointing this out. We improved the graphic design of Figure 3.
- C) Extensiveness of the references of the manuscript is not due to the extensiveness of the published works of plasma- treated liquid in the dentistry field but of a review not focused enough on the field. As reader, I expect references more focused on dentistry applications of NTAPPs/CAPs.
Response: Thanks for pointing this out. We found a new reference about the effect of plasma-activated water (PAW)intake on the mineral composition and surface features of mouse teeth and also its oral toxicity evaluation, that was inserted in the first paragraph of the section 3.3 (Anti-inflammatory properties and wound healing). We have also added a new reference about the effect of plasma activated-liquids (PAL) on an oral squamous cell line in the last paragraph of the section 3.4 (Anti-cancer therapy). Now, we believe that all the extensiveness of references about PAW/PAL related to dentistry were inserted in the manuscritp. If you know more manuscripts about PAW/PAL related to dentistry, please let we know the reference(s) that we will add in the manuscript with great pleasure. Considering that you also expected more references focused on dentistry related to direct applications of NTAPPs,in the sections 3.2 (treatment of oral infections diseases) and 3.3 (Anti-inflammatory properties and wound healing) were inserted more references/informations about NTAPP in dentistry. The role of NF-κB e STAT3 signaling pathways in dentistry was also explained considering that PAW/ PAL may act in these pathways in some cells. Additionally, the effects of NTAPP and PAW/PAL in non-oral cells and tissues were shortened in the sections 3.3 and 3.4, in order to make the manuscript more focused on dentistry. All the additions related to dentistry are highlighted in yellow in the text.
D)Introduced abbreviations in “Introduction” or Part 2. are once again defined in Part 3 (eg: CAP, l. 33 & 270). Moreover, it would be more accurate to include in the introduction the entire paragraph (l. 270) into the subsection “3. PAW Applied in Dentistry - 3.1. Decontamination of dental devices” regarding the introduction/definition of Cold Atmospheric Plasmas: “Cold atmospheric plasma (CAP) is a safe technology that uses low temperature without the generation of toxic residues and chemical products [68]. CAP present a harmful effect to different microbial species [72–74]. Charged particles, emission of UV radiation and specially the reactive oxygen and nitrogen species, generated by CAP, are considered the contributing factors for its antimicrobial property [75]. In addition to being useful in microbial decontamination process”.
Including some aspects are repeated from the introduction (l.77): “NTAPPs, it is important to know that it is an ionized gas consisting of electrons, ions, metastable species, ultraviolet (UV), visible (VIS) radiation, electromagnetic field, and reactive species”.
These details, among others, revealed a lack of coordination/interconnection in the writing of the different parts pf the manuscript.
(Personal opinion: The authors should also consider introducing/defining NTAPPs and CAPs at the beginning of the paper and use a unique abbreviation along the manuscript)
Response: Thanks for pointing this out. We made changes in order to improve the coordination/interconnection in different parts of the manuscript. Repetitive statements such as the one mentioned above were removed from the section 3.1 (Decontamination of dental devices). Abbreviations were checked and only introduced in the introduction or at first mention in the main text. Additionally, we have replaced all CAPs abbreviations to NTAPPs along the manuscript.
E)The authors should consider reducing the 2-pages section on “3.4. Anticancer therapies” (L 570) into the section “PAW Applied in Dentistry” and focusing just on oral cancers, especially on the neck squamous cell carcinoma (SCC), the main one (as correctly stated). However, this section includes extensive parts regarding the use of CAPs/NTAPPs and its proved effectiveness onto a broad range of kinds of cancers, that could be reduced to a simple paragraph with the adequate references.
Response:Thanks for pointing this out. The effects of PAW/PAL and specially NTAPP in non-oral cells and tissues were shortened in the section 3.4 (Anti-cancer therapy), in order to make the manuscript more focused on dentistry. Additionally, we have also added a new reference about the effect of PAL on an oral squamous cell line in the last paragraph of the section 3.4 (Anti-cancer therapy).
- F) *** Details: “RONs” -> “RONS” (l. 186)
Response:Thanks for pointing this out. This error was fixed.
*** In “3.4. Anticancer therapies”: “promising source (l. 584), … promising responses (l. 589) promising results (l. 592), “…is promising considering that”… (l. 610)”: Try to not overused the term “promising” and state/detail the important results of the cited reference regarding the promising aspects.
Response:Thanks for pointing this out. We have reviewed all the manuscript. Some “promising” results and responses were removed and the references better explained, as you requested. Other ones were replaced by suitable terms. Now the term “promising” is not overused (it was inserted only three times in all the manuscript).
The manuscript needs revision, work of synthesis, coordination of writing between the different sections, improvement of the clarity and moreover focusing on dentistry applications to bring originality with respect to the existing reviews on plasma-treated liquids for biomedical applications.
Response:Thanks for pointing this out. We have performed modifications to improve all these issues, as explained in the responses above.

Reviewer 3 Report
This manuscript provide a review of Applications of Plasma-Activated Water in Dentistry. The aim of this paper is to discuss the applicabil-20 ity of PAW in different areas of dentistry according to the published literature about NTAPPs and 21 plasma-liquid technology. Here is my comments:
- The title is applications of Plasma-Activated Water in Dentistry. I don't understand what the relationship between the section of 3.4. Anticancer therapy and dentistry.
- several important areas of Dentistry, such as in Cariology, Periodontology and Endodontics were not even mentioned in this review.
- For the section of 2.1 main reactors of producing, please show the schemes of apparatus used in PAW generation.
Author Response
RESPONSE TO REFEREES: IJMS - 1647046 entitled " Applications of Plasma-Activated Water in Dentistry: A Review"
We appreciate the time and effort that the reviewers have dedicated to providing their valuable feedback on our manuscript. We are grateful to the reviewers for their insightful comments on our paper. We have been able to incorporate changes to reflect most of the suggestions provided by the reviewers. We have highlighted the revisions within the manuscript.
Below is a point-by-point response to the reviewers’ comments and concerns.
Reviewer Comments, Author Responses, and Manuscript Changes
REVIEWER 3
This manuscript provide a review of Applications of Plasma-Activated Water in Dentistry. The aim of this paper is to discuss the applicabil-20 ity of PAW in different areas of dentistry according to the published literature about NTAPPs and 21 plasma-liquid technology. Here is my comments:
Comment 1:
1) The title is applications of Plasma-Activated Water in Dentistry. I don't understand what the relationship between the section of 3.4. Anticancer therapy and dentistry.
Response 1:Thanks for pointing this out. In the mouth several types of cancer can be observed. The most prevalent (>90%) is the squamous cell carcinoma (SCC), which can affect any intraoral site but is more common on the tongue. The dentists have the role of diagnosing oral cancers by clinical exams and biopsies and after following these patients during the treatment that may include surgery, chemotherapy and/or radiotherapy. Thus, anticancer therapy is also an important topic in dentistry. Many researchs study complementary therapies for the oral SCC. NTAPPs is being studied as a possible complementary therapy, with interesting results in advanced cases of oral SCC. The prevalence of oral SCC in the mouth as well as the role of NTAPPs as a complementary therapy, and also the perspectives for plama-activated water (PAW) in this area are discussed in the section 3.4 (Anticancer therapy), as follows:
3.4. Anti-cancer therapy
Sensitivity of cancer cells to NTAPPs has been demonstrated in many studies. Reactive oxygen and nitrogen species may penetrate cancer cells easier compared to health ones, make them more vulnerable to their harmful effects. This fact may be explained by the higher amount of water channels (aquaporins) in cancer cells, that facilitates the transport of reactive species into cytosol. Additionally, the lipid peroxidation caused by free radicals generates pores in the membrane, which also allow the entry of reactive species into the cell. This process is attenuated by the condesation of membrane lipids in normal cells that are rich in cholesterol. However, cancer cells usually present fewer amount of lipids, which impairs this defense mechanism. The large influx of reactive species into cancer cells triggers signaling cascate pathways that may culminate in different types of cell death, such as apoptosis, necrosis or senescene, depending on the dose of exposure. Another important anti-cancer molecular mechanism of NTAPPs is its capacity in reducing the expression of some integrins. These molecules are essential for adhesion, migration, and invasion of cancer, which indicates that NTAPPs can be useful against metastases [31].
Besides the direct action of NTAPPs in cancer cells, it may be usefull in this approach by its interaction with the tumor microenvironment. Reactive oxygen and nitrogen species are able to damage important extracellular matrix components, such as collagen, fibronectin and hyaluronic acid. The induction of an antitumor immunity have also been proposed as an action mechanism [31]. Anti-cancer properties of NTAPPs have been observed against many types of cancers cells [156–160]. Interestingly, clinical reports have already been conducted showing the role of NTAPPs in advanced squamous cell carcinoma (SCC) [154,161,162], most of them in intraoral sites [154,161]. SCC is the most prevalent oral cancer. More than 90% of the cases occur in man over 45-year-age exposed to tobacco and/or alcohol. The lip is the most prevalent site, followed by the tongue [163]. An improvement in the quality of life of patients with advanced SCC located at intra-oral sites or jaws was described after NTAPPs treatment, by the reduction of odor and pain. Partial remission of the lesion occured in some cases [154,161]. Additionally, reduction of microbial load, wound healing of some infected ulcerated areas [154], and enhance of apoptotic cells were described [162]. Partial or total remission of pre-malignant lesions of skin resulting from chronic ultraviolet exposure, referred as actinic keratoses, were also observed after NTAPP treatment [164], which opens perspectives for actinic cheilitis, the pre-malignant lip counterparty, that precedes the emergence of lip SCC [165]. The adjunct treatment of initial SCC with NTAPPs has not been evaluated yet and it would be interesting considering some in vitro responses of oral SCC to NTAPPs. Synergic effect of cisplastin and NTAPP against oral SCC cells in vitro was described associated with low cytotoxicity to normal oral cells [166]. Moreover, combination of NTAPP with cetuximab inhibited invasion/migration of cetuximab-resistant oral SCC cells in vitro [167].
The possibility of using this techonlogy through plasma activated liquid is promising considering that PAL/PAW may be injected into large or deep tumors, facilitating the action in the entire lesion. Moreover, this kind of treatment probably would be faster and easier for the clinician compared to NTAPPs and more confortable for the patient who usually is weakened by radiotherapy and/or chemotherapy. The treatment of liquids directly on substrate or indirectly (later in contact with the substrate) has been perfomed satisfactorily in many types of cancer cells with the use of different liquids, such as deionized water, cell culture media, Ringer’s solution and saline [17,18,168–172]. Apoptotic cells were observed in cancer cells exposed to activated-deionized (DI) water [18,172]. Different studies have shown that PAL are also efficient against cancer, due to toxic effects of oxygen and nitrogen species accumulated in these solutions and also by immuno-stimulatory properties [173]. Different cancer cells may respond to PAL with decreased proliferation and migration and increased cell death by apoptosis, necrosis, autophagy and senescence [174]. Reduction of tumor burden and metastasis-inhibitory effect were also observed with the use of PAL [170]. It was suggested that RNS could play a more relevant role in cancer cells death than ROS [172].
A previous study evaluating the effect of PAL on an oral squamous cell line (SCC15) observed anti-cancer capacity of the plasma-activated medium. Reduction in cell viability was observed with increasing incubation time. Moreover, they have demonstrated that many signaling pathways, such as p-53 pathway, could play a critical role in this process [175]. The effectiveness of PAW, as well as its possible mechanisms of action in oral cancer, has not been evaluated yet. Considering that the treatment of SCC is the entire remotion of the lesion, this kind of treatment could be useful in two situations: 1) Prior the surgery, by washing the lesion or through the injection of plasma activated water in deep neoplasms, and 2) After the surgery, by reducing the microbial load and favoring the wound healing. The anticancer properties could also be positive to avoid recurrences. The role of plasma-activated water in oral premalignant lesions, such as oral leukoplakia, erythroplasia and actinic cheilitis should also be evaluated. Different kinds of treatments have been used in patients with actinic cheilitis, such as combinatory treatment with PDT and laser ablation. However, carbon dioxide laser ablation and vermilionectomy, that are invasive for the patients, have been considered the most effective treatments [165]. In this way, plasma-activated water could represent a non-invasive approach to be used in oral premalignant lesions with other therapies or even alone, depending on the results and risk factors of each patient.
Comment 2:
- several important areas of Dentistry, such as in Cariology, Periodontology and Endodontics were not even mentioned in this review.
Response 2:Thanks for pointing this out. These areas are really very important for dentistry. In the section 3.2 (treatment of oral infections diseases) we discussed about dental caries (cariology) from the second to the sixth paragraph. In these paragraphs we explained the etiopathogenesis of dental caries, and the role of NTAPPs and mainly plasma activated-water or liquids (PAW/PAL) as antimicrobial agents against microorganisms that favor the development of dental caries. In this same section in the seventh paragraph we discussed the evolution of dental caries to pulpar infection leading to periapical disease (endodontic infection). The role of PAW against microoganisms that are important in the endodontic infection were also discussed in this paragraph. Posteriorly, in the eighth paragraph we explained the etiopathogenesis of periodontal disease/periodontitis (Periodontology area) and the role of NTAPPs/PAW against the microorganisms that contribute to the development of periodontal disease. The possible advantages of using PAW instead of NTAPPs were mentioned in the discussion of all the cited oral infections diseases. Considering that periodontitis is also an inflammatory disease, studies about NTAPP and periodontitis regarding the anti-inflammatory effect were also discussed in the section 3.3 (Anti-inflammatory properties and wound healing). Additionally, the role of NTAPPs/PAW in other oral diseases such as oral candidiasis and oral lichen planus were pointed in the sections 3.2 + 3.3 and 3.3, respectively. The section 3.2 (treatment of oral infections diseases) and 3.3 (Anti-inflammatory properties and wound healing) may be observed bellow:
3.2. Treatment of oral infectious diseases
The use of PAW in the treatment of oral infectious diseases is a promising field of investigation. Several groups of microorganisms are involved in the etiopathogenesis and progression of the main infectious diseases that affect the oral cavity, such as caries, periodontitis and candidiasis. Considering the antimicrobial properties of NTAPPs, the control of these microorganisms by PAW or other plasma-activated solutions has also been investigated as an alternative treatment to traditional therapies that have their limitations.
Biological, behavioral, psychosocial, and environmental factors are related to the development of caries [106]. The imbalance between demineralization and remineralization, caused by fluctuations in pH, promotes the tooth decay [107]. The metabolic activity of dental biofilm, formed by microorganisms embedded in a matrix of extracellular polymeric substances adhered to the teeth surface, is responsible for these pH fluctuations, especially with the intake of rich sugar diet. The biofilm continues to grow progressively if undisturbed. In its composition there are many groups of microorganisms, and some of them are especially involved in the carious process [108].
Streptococcus mutansand Lactobacillus spp. are considered the main cariogenic bacteria, responsible to produce acid and consequently the demineralization of the tooth structure [109,110]. Interestling, plasma activated liquids (PALs) have shown to be effective against cariogenic microorganisms [22,111]. A previous study observed that PBS or saline solution activated with non-thermal plasma of argon (Ar) and oxygen (O2) for 5 min are able to reduce the number S. mutansviable cells. This reduction was verified in both planctonic and biofilm forms of S. mutans, after 1h of treatment. The authors further reported that the activated liquids were not cytotoxic to fibroblasts [111].
Modifications in the composition of oral biofilm may be observed due to dietary habits, type of dentition (primary or secondary), and even with the disease progression. Although S. mutansand Lactobacillus spp. are considered the most important microogasnisms retated to dental carie, previous findings demonstrated that Actinomyces spp. may be associated with the disease progression in root caries [108]. A previous study evaluated the role of distilled water activated by a plasma jet of Ar and O2 (98% and 2%, v/v, respectively) for 20 min in the reduction of S. mutans, Porphyromonas gingivalisand Actinomyces viscosus. The treatment that was performed from 0 to 120 s reduced the viability of all microbial species. Significant reduction of A. viscosuswas observed within 40 s while S. mutansachieved a similar reduction after 60 s of treatment [22].
Candida albicansis another microorganism that may be found in carious dentin of active root carious lesions and some authors have suggested that this fungus might play a role in the progression of the disease [112]. This species was detected on biofilm in cases of childhood caries [113] and it was associated with an increase of plaque glucosyltransferase (Gtf) enzyme activity, a virulence factor associated with caries, in children with early carious lesions [114]. In addition to its possible role in dental caries, C. albicansis the main species related to oral candidiasis, the most common oral fungal disease. Local and systemic factors such as impaired salivary gland function, inhaled steroids, dentures, oral cancer/leukoplakia, broad-spectrum antibiotics, immunosuppressive drugs and conditions, nutritional deficiencies, diabetes, smoking and Cushing's syndrome may contribute to oral candiadisis [115]. Previous studies demonstrated that the direct application of NTAPPs is effective against C. albicans[116,117], both in planctonic and biofilm forms. Investigations on the effects of PAW on C. albicanshave also been conduced with different methodologies and findings. Reduction of C. albicansviability after 5 min of treatment with distilled water, activated by dielectric barrier discharge (DBD) with atmospheric air, for 5 and 10 min, was reported. The authors attributed the antimicrobial action to the higher concentration of NO-3and lower concentrations of NO-2and H2O2in PAW [118]. Antimicrobial effects against C. albicanswas also reported in hydrogels constituted by plasma activated deionized water, after 24, 48 and 72 h of contact time. An increase of inhibition zones of the microorganism was observed in longer exposure times, with the best result after 30 min. Hydroxyl radical and NO-3were suggested to be the main components with antifungal activity [119]. On the contrary, one study reported no effect of plasma-activated tap water for 10 and 30 min on C. albicans planktonic cells. The water was activated with an atmospheric air plasma, generated by a forward vortex flow reactor (FVFR), for 30 min [16].
In the context of cariology it is also important to emphasize the positive property of NTAPPs in adhesive restorations. Plasma exposure generates the deposition of free radicals and ions on tooth substrate changing the surface proteins of dentin what has led to increased bond strength in adhesive restorations. The enhance in bone bond strength avoid microleakage and consequently prevents secondary caries [40]. It is not known whether the PAW can also increase the bond strength improving the restoration performance. Probably, this process could also occur with plasma-liquid technology due to the action of the reactive species on the surface of dentin.
The evolution of carious process, traumatic injuries and cracks allow that pathogens and their products passing through the dentin and reach the pulp. The pulpal infection frequently progress to necrosis of the tissue and the infection may spread to the apex of the tooth promoting periapical disease [120]. Enterococcus faecalisis commonly found in primary and secundary endondontic infections [120–122]. This microoganism express virulence factors and resistance mechanisms that favor its presence in the root canal and consequently can lead to endodontic therapy failure [120,123]. Some studies have investigated the action of PAW on E. faecalis[99,124], which is interesting for dentistry once it could be used as an antimicrobial irrigator in endodontic treatments. Considering the particularities of root canals, such as the presence of accessory canals, the antimicrobial irrigation with PAW would be more useful than the direct application of NTAPPs. As mentioned in the last topic, reduction of E. faecalisviable cells in 5-day DUWLs biofilms was observed after 1-3 min of treatment with preoviously activated water. The treatment of the E. faecalissuspensions, in deionized water, was also performed with satisfactory results. The bacterial suspension was exposed to microjet plasma formed by atmospheric air for 10 to 90 s. The antimicrobial effect occurred progressively after 45, 60 and 90 min of treatment, while inhibitory effects on biofilm formation was detected even in shorter exposure times (10, 20 and 30s) [124]. Another microorganism that may be isolated from root canals, though not often, is Escherichia coli[120]. It was demonstrated that the planktonic treatment with PAW significantly reduced the number of E. colicolonies after 10 min of exposure [16]. In this same study, 10 and 30 min of PAW contact was effective against Staphylococcus aureus, a bacteria commonly found in chronic osteomyelitis of jaws in association with anaerobic pathogens [125].
The development and progression of periodontal disease are related to specific groups of Gram-negative bacteria, the so-called periodontopathogens. The transition from health periodontium to periodontitis is related to three important factors: the polymicrobial synergy, the dysbiotic microbiota, and a susceptible host [126]. The multifactorial etiology of periodontal disease contributes to the difficulty in the treatment, and alternative therapies have been investigated [127]. Porphyromonas gingivalis, Treponema denticola, Tannerella forsythia, and Aggregatibacter actinomycetemcomitanshave been considered important periodontopathogens [128]. A progressive inhibition of P. gingivalisin planktonic and biofilms forms was previously observed after 1, 3, 5 or 7 min of NTAPP exposure. Moreover, improved periodontal tissue recovery was obtained after 5 min of NTAPP exposure, proportionally to the number of applications [129]. Insterestling, a previous study demonstrated that PAW also may be effective against P. gingivalis. A reduction of 5-log in planctonic bacteria was observed after 20 s of exposure [22]. Considering the particularities of subgingival biofilm, and its relationship with the progression of the disease, the treatment with PAW would be even more interesting since it would be able to reach areas of restricted access, such as the subgengival sites.
The mechanisms of action suggested in all of these studies evaluating the antimicrobial properties of PAW mainly involved the reactive oxygen and nitrogen species (RONS) produced in the solutions by plasma activation. Different biological effects could be observed in each study, which probably is related to differences in methodology and group of microorganism. Further in vitro and in vivo investigations are needed to standardize the best parameters for each solution and microorganism. Considering the pontential antimicrobial effects observed in these studies, they probably could contribute to the treatment of oral infectious diseases.
3.3. Anti-inflammatory properties and wound healing
An important feature of non-thermal plasma is the possibility of tissue antisepsis without causing damage, which makes NTAPPs a good alternative treatment for infections diseases. This selectivity, probably occurs due to biochemical, metabolic and cell cycle differences between eukaryotes and prokaryotes and also surface/volume ratio of mammalian cells, that is higher compared to bacterial and fungal ones [130]. It was demonstrated that no important side effect occurs after oral application of NTAPP on the mucosa of mice, in short-term experiments [131]. A previous study investigated the effect of PAW intake in mice after 90 days administration once its use in dentistry may lead to accidental ingestion. The mineral composition and surface micro-morphology of vital mouse teeth after long-term exposure, as well as local and sistemic toxicity were evaluated. The authors observed that there were not significant changes in the mineral composition and surface micro-morphology of the teeth. Moreover, the long-term exposure was not toxic to tongue, oral mucosa, sublingual glands or other body organs, which presented normal structure and physiology [132]. In addition to not being harmful to mammalian cells, NTAPPs have demonstrated to decrease inflammation and contribute to tissue repair [133].
Studies on skin inflammatory diseases such as allergic contact dermatitis and atopic dermatitis have shown anti-inflammatory effects of non-thermal plasma [134–136]. These effects have also been observed in oral studies. The treatment of oral candidiasis in mice showed low occurrence of inflammatory alterations. After plasma exposure, the cell inflammatory infiltrate was predominantly mononuclear and macrophage-rich, with scarce polymorphonuclear cells [116]. Additionally, an study evaluating the role of NTAPP as an adjuvant therapy for the treatment of periodontitis induced in rats observed that the expression of inflammatory-related cytokines such as TNF-α and IL-1β decreased significantly in the group where NTAPP was used as an adjuvant approach, while the level of the anti-inflammatory cytokine IL-10 showed a significant increase [137].
The behavior mast cells and keratinocyte cell line (HaCat) after the contact with non-thermal plasma-activated medium has been analyzed. Interestingly, the plasma-activated liquid prevented an enhance of the pro-inflammatory genes and cytokines TNF-α, IL-6 and IL-13 in activated mast cells, by inhibiting the NF-κB signaling pathway. The activation of NF-κB by TNF-α/IFN-γ was also inhibited in HaCat cells suggesting that this treatment could be effective against, not only acute, but also chronic inflammation [136]. Similarly, in another study the pro-inflammatory responses of HaCat activated by TNF-α/IFN-γ or LPS was also supressed by PAL. Moreover, STAT3, which is an important pathway for Th17 cell activation, was inhibed by PAL in IL-6-stimulated HaCaT [135]. In this way, plasma activated liquids have shown to act in different inflammatory signaling pathways of keratinocytes. These findings are important for dentistry once NF-κB e STAT3 pathways are involved in the etiopathogenesis of periodontitis [138]. Moreover, these pathways also play a role in oral candidiasis once mucosal candidiasis promotes NF-κB activation [139] and STAT3 signaling are related to IL-17-mediated immunity to oral mucosal candidiasis [140] .
The possibility of treating autoimmune skin diseases by direct application of plasma or by plasma activated-liquids may also be interesting for dentists which also have to deal with some autoimmune conditions, such as oral lichen planus [141] pemphigus and mucous membrane pemphigoid [142]. The effects of NTAPP on oral lichen planus (OLP) was previously investigated [143]. For this, biopsies from healthy and OLP areas were performed followed by the application of NTAPP in the ex-vivo tissues for 3 min. From these lesions, 24 were reticular, 3 erosive, and 1 atrophic. The treatment decreased the infiltration of T-cell in OLP compared with healthy samples. Additionally, the levels of IL1β, IL2, IL10, and GM-CSF decreased significantly after the treatment, and a tendency to decrease other inflammatory markers was observed, suggesting an immunomodulatory role of NTAPPs in OLP. The authors also presented a clinical report from a 73-year-old man suffering from an erosive OLP. The treatment consisted of 5 min of application, 2 to 3 times per week (12 sessions). It promoted relief of the burning sensation, after 4 sessions. During the treatment, the oral inflammation decreased and the ulcerated area of the lesion healed almost totally [143]. Considering that the erosive presentation is usually symptomatic, requering the treatment with topic steroids and sometimes systemic ones [141], clinical studies evaluating a large number of these cases are welcome. The efficiency of PAW should also be evaluated, since the washing with a plasma-activated liquid could reach the entire are of OLP without the need of direct application in several points of the lesion.
The exposure of diabetic animals to NTAPPs have also shown anti-inflammatory properties [144,145] and improvement in the wound healing [146], which brings perspectives for dentistry, especially considering the proved relationship between diabetes and periodontal disease. Severe periodontal destruction is usually observed in diabetic patients while poor glycemic control in more common in diabetic people who also have periodontal disease [147]. Interestingly, diabetic mice treated with NTAPP showed a decrease of oxidative stress biomarkers, advanced glycation end products (AGEs) and inflammatory cytokines, such as IL-1, IL-6, and TNF-α [145]. It has been suggested that increased accumulation of AGEs and their interaction with specific receptors (RAGE) in diabetic gingival tissue could promote the hyperproduction of proinflammatory cytokines, as well as vascular alterations and loss of tissue integrity, contributing to the worsening of periodontitis [148]. Thus, the adjunct treatment of diabetes mellitus with plasma modalities possible could act indirectly and positively in periodontitis and in other inflammatory oral diseases. The direct action of NTAPPs or PAW in periodontal disease should also be considered, since it is a multifactorial condition in which the microbial biofilm activates the immune system with production of proinflammatory cytokines and consequenty tissue loss [149]. The selectivity role of plasma could be useful in both, in the elimination of periodontopathogenic microorganisms and also in the gingival tissue, decreasing the inflammatory process. Considering the generalized subtype of chronic periodontitis, the treatment with PAW could be clinically easier reaching different affected areas in a single use.
The anti-inflammatory and microbicide function of NTAPPs have been demonstrated to favor the tissue repair [133]. Clinical trials evaluating NTAPPs have already been performed in which this technique was considered safe, painless and effective against bacterial load [150,151]. Solutions activated with plasma, such as medium, saline and water have also shown good results in vitro and in vivo concerning the wound healing [19,133,152]. Cell proliferation and migration were observed in human keratinocytes exposed to 15 seconds of medium activated with Helium and Argon (He/Ar)-generated NTAPP [133], which could favor the re-epithelialization of wounds on the skin and also on the oral mucosa that has keratinocytes in the epithelial composition. There is no study evaluating the effect of PAW or PAL on mouth wound healing although two studies carried out in rats and mice have observed a tendency to improve periodontal tissue loss after the treatment of experimental peridontitis with NTAPPs [137,153]. Moreover, wound healing of some infected ulcerated areas of advanced oral squamous cell carcinoma was observed after NTAPP exposure[154]. Considering that wound repair is important for different modalities of dentistry such as oral surgery, periodontics, oral pathology and implantology, PAW could be an option to accelerate the healing after oral diseases or oral surgeries.
NTAPPs were previously suggested as a good possibility for oral surgery because NTAPP was tested with osteoblast-like cells (MG-63) leading to cell proliferation and in vitro wound closure [155]. The oral implant modification with NTAPPs has also been suggested since this treatment may enhance the roughness and wettability of the implant surface thus improving the cell adhesion and consequently the osseointegration [40]. These results with direct application of NTAPPs open perspectives for the use of PAW in oral surgery and implantodology. PAW could be used even more easily in these procedures, such as in surgeries for the remotion of oral lesions and tooth extractions, especially in impacted third molars. Thus, the use of PAW in dentistry should be considered given the antimicrobial, anti-inflammatory and wound healing properties of NTAPPs, in addition to their ability to alter the surface of dental implants. The simplicity of the technique, considering the use of PAW as a mouthwash or an irrigation agent, and possibly the lower price of PAW compared to direct application of NTAPPs, which would necessarily demand a device in the dental office, make PAW a potential adjuvant oral tool for conditions requiring tissue repair.
Comment 3:
- For the section of 2.1 main reactors of producing, please show the schemes of apparatus used in PAW generation.
Response 3:Thanks for pointing this out. We have rewritten section 2.1 to improve understanding and to avoid confusion. To clarify these points, we added Figure 2 and the following paragraphs between lines 148 and 181.
“It is important to note that NTAPPs operate at high voltage. As reported in the literature, these voltages range from 1 to 50 kV, with operating frequencies that can start in tens to thousands of Hz (kHz) and powers that generally do not exceed values greater than 10 W. Commonly used working gases are helium (He), argon (Ar), oxygen (O2), nitrogen (N2), air, or a mixture of these gases. Working gases are used with flow rates ranging from 1 to 30 L/min [X18-X25]. Indeed, there is a range of reactors used for water and liquid activation, and every day a new article is published with new reactors that have minor changes. Therefore, a dedicated review article would be needed to demonstrate the NTAPP generation reactors and their characteristics, but the focus of the present work is not that. Thus, below we show some reactors used to generate PAW.
(See figure 2 in the manuscript)
As seen in Figure 2, discharges are used directly into the water and on the surface of the water, which considerably affects the chemical composition of the PAW. This difference in chemical composition is basically due to the differences between the rupture forces in the gas phase (discharge on the surface) and the water (discharge inside) [X, X]. However, as reported in the literature, the most applied systems are those that operate with plasma discharge in contact with water, i.e., PJ and DBD as a plasma source [X, X]. As demonstrated in the next section, these plasmas deliver the RONS from the plasma gas to the liquid phase more efficiently.
An important question is whether there is commercial equipment that can be applied in clinical practice. Recently, Andrasch et al. [X], Pemen et al. [X], and Schnabel et al. [X] developed pilot units with potential for practical applications. Andrasch et al. [X] and Schnabel et al. [X] obtained a PAW production rate of 1L/min. Pemen et al. [X] obtained 0.5 L/min. However, these devices cannot meet the application requirements in dentistry and medicine, which are low pH and high concentrations of RONS. On the other hand, in agriculture and in the food industry, the production of millions of liters of PAW at a low cost is required. Therefore, it can be said that PAW generating equipment is still in the pilot phase and has great commercialization potential in the coming years.”
Round 2
Reviewer 1 Report
The Authors have done a great work to improve the manuscript. Аll comments of the reviewer were answered thoroughly. Thanks the Researchers for an interesting paper.
Author Response
We thank the reviewer for the comments provided.
Reviewer 2 Report
Details:
L 184 – “2.2. The origin reactive oxygen and nitrogen species”, missing “of” in the subtitle section. “2.2. Origin of reactive oxygen and nitrogen species”
L 202 – I would suggest modifying the sentence “In Appendix A, the generation mechanisms of the main plasma-induced long-lived reactive species in the water activation process that play a crucial role in healthy applications are described.” By: In Appendix A, the generation and recombination mechanisms of the main plasma-induced long-lived reactive species described in water activation processes that play a crucial role in healthy applications.”
L 205 - Figure 3 - The authors should consider modifying the title “Reactive species induced at gas phase, plasma phase and plasma-water interaction in NTAPP”, in the top part of the figure by something like “Reactive species induced IN gas phase, plasma phase, plasma-water interface and liquid phase in NTAPP”. They should also consider the modification of the “green” block Plasma-water interaction by “Liquid phase.” Indeed, some species (ex. Peroxynitrites) does not come from the direct interaction plasma/water but from a recombination of the reactive species generated by NTAPP.
L 244 – “−80 °C [96,97]. However, the antimicrobial activity may be reduced depending on to the storage temperature, by decreasing the number of reactive species. Inactivation of S. aureus was observed after 20 min of treatment with PAW stored at -80°C. However, this potential decreased significantly when plasma-activated distilled water was stored at -20 °C for 1, 7, 15 and 30 days [96]. Similarly, another study observed that 60 min of exposure to PAW stored at –80 or –150°C” The authors should harmonize the space (or no space) between numbers and units. Check this along the complete manuscript.
L 657 - Appendix A 656 “In this section we will show the main mechanisms for generating long-term RONS 657 that are found in PAW after the plasma-liquid phase interaction. Hydrogen peroxide 658 (H2O2), nitrite (NO2), nitrate (NO3-), nitrous acid (HNO2) and ozone (O3) are the main 659 reactive species that will be presented below.” – Subindex format of the RONS missing.
References: [26] & [183] repeated/same reference. Change [183] by [26] in the manuscript.
Author Response
RESPONSE TO REFEREES – R2: IJMS - 1647046 entitled " Applications of Plasma-Activated Water in Dentistry: A Review"
We appreciate the time and effort that the reviewers have dedicated to providing their valuable feedback on our manuscript. We are grateful to the reviewers for their insightful comments on our paper. We have been able to incorporate changes to reflect most of the suggestions provided by the reviewers. We have highlighted the revisions within the manuscript.
Below is a point-by-point response to the reviewers’ comments and concerns.
Comments and Suggestions for Authors
Details:
Comment 1:L 184 – “2.2. The origin reactive oxygen and nitrogen species”, missing “of” in the subtitle section. “2.2. Origin of reactive oxygen and nitrogen species”
Response 1: Thanks for pointing out this error. We fixed this error.
Comment 2:L 202 – I would suggest modifying the sentence “In Appendix A, the generation mechanisms of the main plasma-induced long-lived reactive species in the water activation process that play a crucial role in healthy applications are described.” By: In Appendix A, the generation and recombination mechanisms of the main plasma-induced long-lived reactive species described in water activation processes that play a crucial role in healthy applications.”
Response 2: Thanks for pointing this out. We modified the sentence according to your suggestion.
Comment 3:L 205 - Figure 3 - The authors should consider modifying the title “Reactive species induced at gas phase, plasma phase and plasma-water interaction in NTAPP”, in the top part of the figure by something like “Reactive species induced IN gas phase, plasma phase, plasma-water interface and liquid phase in NTAPP”. They should also consider the modification of the “green” block Plasma-water interaction by “Liquid phase.” Indeed, some species (ex. Peroxynitrites) does not come from the direct interaction plasma/water but from a recombination of the reactive species generated by NTAPP.
Response 3: Thanks for pointing this out. We improved Figure 3 and the caption too.
Comment 4:L 244 – “−80 °C [96,97]. However, the antimicrobial activity may be reduced depending on to the storage temperature, by decreasing the number of reactive species. Inactivation of S. aureus was observed after 20 min of treatment with PAW stored at -80°C. However, this potential decreased significantly when plasma-activated distilled water was stored at -20 °C for 1, 7, 15 and 30 days [96]. Similarly, another study observed that 60 min of exposure to PAW stored at –80 or –150°C” The authors should harmonize the space (or no space) between numbers and units. Check this along the complete manuscript.
Response 4: Thanks for pointing this out. We harmonize the spaces between numbers and units.
Comment 5:L 657 - Appendix A 656 “In this section we will show the main mechanisms for generating long-term RONS 657 that are found in PAW after the plasma-liquid phase interaction. Hydrogen peroxide 658 (H2O2), nitrite (NO2), nitrate (NO3-), nitrous acid (HNO2) and ozone (O3) are the main 659 reactive species that will be presented below.” – Subindex format of the RONS missing.
Response 5:Thanks for pointing out this error. We fixed this error.
Comment 6:References: [26] & [183] repeated/same reference. Change [183] by [26] in the manuscript.
Response 6:Thanks for pointing out this error. We fixed this error.
Reviewer 3 Report
The manuscript could be accepted
Author Response

(The authors gave the same response as above.)
